



# MODIS Daily Cloud-gap-filled Fractional Snow Cover Dataset of the Asian Water Tower Region (2000-2022)

Fangbo Pan[1], Lingmei Jiang[1, *], Gongxue Wang[2], Jinmei Pan[3], Jinyu Huang[1], Cheng Zhang[1], Huizhen Cui[1], Jianwei Yang[1], Zhaojun Zheng[4], Shengli Wu[4], Jiancheng Shi[5]

[1]State Key Laboratory of Remote Sensing Science, Jointly Sponsored by Beijing Normal University and Aerospace Information Research Institute of Chinese Academy of Sciences, Faculty of Geographical Science, Beijing Normal University, Beijing 100875, China

[2] College of Geomatics, Xi'an University of Science and Technology, Xi'an 710054, China

[3] State Key Laboratory of Remote Sensing Science, Aerospace Information Research Institute, Chinese Academy of Sciences, Beijing 100101, China.

[4] Satellite Meteorological Institute, National Satellite Meteorological Center, China Meteorological Administration, Beijing 100081, China

[5] National Space Science Center, Chinese Academy of Sciences, Beijing 100190, China

*Correspondence to*: Lingmei Jiang (jiang@bnu.edu.cn)

**Abstract.** Accurate long-term daily Cloud-gap-filled fractional snow cover products are essential for climate change and snow hydrological studies in the Asia Water Tower (AWT) region, but existing Moderate Resolution Imaging Spectroradiometer (MODIS) snow cover products are not sufficient. In this study, the multiple endmember spectral mixture analysis algorithm based on automatic endmember extraction (MESMA-AGE) and the multistep spatiotemporal interpolation algorithm (MSTI) are used to produce the MODIS daily cloud-gap-filled fractional snow cover product over the AWT region (AWT MODIS FSC). The AWT MODIS FSC product have a spatial resolution of 0.005°, and spans from 2000 to 2022. The 2745 scenes of Landsat-8 images are used for the areal scale accuracy assessment. The fractional snow cover accuracy metrics, including coefficient of determination ($R^2$), root mean squared error (RMSE), and mean absolute error (MAE) are 0.80, 0.16 and 0.10, respectively. The binarized identification accuracy metrics, including overall accuracy (OA), producer's accuracy (PA), and user's accuracy (UA), are 95.17%, 97.34% and 97.59%, respectively. Snow depth data observed at 175 meteorological stations are used to evaluate accuracy at point scale, yielding the following accuracy metrics: an OA of 93.26%, a PA of 84.41%, a UA of 82.14%, and a cohen's kappa (CK) value of 0.79. Snow depth observations from meteorological stations are also used to assess the fractional snow cover resulting from different weather conditions, with an OA of 95.36% (88.96%), a PA of 87.75% (82.26%), a UA of 86.86% (78.86%) and a CK of 0.84 (0.72) under the MODIS clear sky observations (spatiotemporal reconstruction based on the MSTI algorithm). The AWT MODIS FSC product can provide quantitative spatial distribution information of snowpack for mountain hydrological models, land surface models, and numerical weather prediction in the Asia Water Tower region. This dataset is freely available from the National Tibetan Plateau Data Centre at https://doi.org/10.11888/Cryos.tpdc.272503(Jiang et al., 2022) or from the Zenodo platform at https://zenodo.org/doi/10.5281/zenodo.10005826.



## 1 Introduction

Snow cover has the characteristics of high albedo, low emissivity and strong water-holding energy (Yang et al., 2014; Wang et al., 2022; Pan et al., 2023; Wang et al., 2023). The extent and variability of snow cover have profound implications for global and regional water and energy cycles(Elguindi et al., 2005; Senan et al., 2016) and climate change (Barnett et al., 2005; Li et al., 2018). The Asian Water Tower region centered on the Tibetan Plateau is the region with the largest snow accumulation outside the North Pole and South Pole (Immerzeel et al., 2020). In recent years, this region has been in a state of imbalance,

which is mainly reflected in the massive conversion of snow (one of the two forms of solid water) into liquid water (Yao et al., 2022). Furthermore, since 2008, the snow cover in the Asian Water Tower region has surpassed the tipping point and has become unstable(Liu et al., 2023) , which had a strong and stable relationship with the changes in the Amazon rainforest ecosystem. Therefore, it is important to produce long-term series and high spatiotemporal resolution Cloud-gap-filled fractional snow cover datasets in the Asian Water Tower region.

Remote sensing technology has become an essential tool for monitoring snow cover globally. Polar orbit satellites such as NOAA/AVHRR, Terra/Aqua MODIS, Landsat, and Sentinel-2, which are often used to monitor snow cover, have spatial resolutions ranging from meters to kilometers. Toward the requirement of daily, large-scale, and long-time series of fractional snow cover monitoring, only moderate to coarse resolution sensors, such as AVHRR and MODIS are currently available. However, multispectral images at moderate and coarse spatial resolution have mixed pixels near the snow line, the edge zone

of snow patches and the forest area covered by snow (Painter et al., 2009; Pan et al., 2022; Wang et al., 2022)**.** The classification of snow and non-snow alone will lead to significant overestimation or underestimation. Classification errors will be further transferred the subsequent applications in various fields(Wang et al., 2013; Niittynen et al., 2020; Notarnicola, 2020). The existing optical remote sensing snow cover mapping methods mainly include the reflectivity linear interpolation method(Metsämäki et al., 2012; Metsamaki et al., 2005; Wang et al., 2017) , snow index empirical relationship method(Hall

et al., 1995; Salomonson and Appel, 2004; Wang et al., 2021; Salomonson and Appel, 2006; Wang et al., 2020), machine learning method (Dobreva and Klein, 2011; Czyzowska-Wisniewski et al., 2015; Kuter, 2021; Xiao et al., 2022) and spectral mixture analysis method (Painter et al., 2003, 2009; Bair et al., 2021). The accuracy of the first three methods depends on training data, and the methods need to be retrained when used in different regions and on different dates. The MEAMA-AGE algorithm is a kind of automatic extraction of pure snow and non-snow endmembers based on the single-band reflectance of

MODIS multispectral images and the normalized differential snow index (NDSI), normalized differential vegetation index (NDVI) and normalized differential water index (NDWI), and then fractional snow cover is retrieved by the MESMA-AGE algorithm (Shi, 2012; Zhu and Shi, 2018). This algorithm can ensure the representativeness of the endmember, improve the computational efficiency and effectively adapt to the characteristics of strong topographic heterogeneity and thin and broken snow, with better accuracy and robustness than other algorithms (Hao et al., 2019; Pan et al., 2022).

Terra and Aqua MODIS provide two daily daytime observations, but the MODIS annual average cloud cover in the Asian Water Tower region is proximately 50%(Wang et al., 2019; Huang et al., 2022).  Snow cover observations can be obscured by





clouds, resulting in many data gaps in daily snow cover products, which greatly limits the application of daily snow cover products. To improve the spatiotemporal continuity of snow cover products, researchers have proposed various spatiotemporal reconstruction algorithms, such as temporal methods (Dozier et al., 2008; Tang et al., 2017; Tran et al., 2019), spatial

methods(López-Burgos et al., 2013; Shea et al., 2013; Hou et al., 2019), spatiotemporal methods(Li et al., 2017; Huang et al., 2018; Li et al., 2020; Xing et al., 2022), and multisource data fusion methods(Yang et al., 2014; Yu et al., 2016; Dai et al., 2017). Most existing algorithms were developed for binary snow cover products, and although they have good accuracy, they are difficult to apply to continuous values such as fractional snow cover. The multistep grouping algorithm used in this study is an improved spatio-temporal method that combines spatial and temporal methods through multiple step implementations

(Parajka and BlöSchl, 2008; Gafurov and Bárdossy, 2009; López-Burgos et al., 2013). These simple multistep combinations have been shown to be effective and efficient in cloud removal, and agree very well with *in situ* observations (Paudel and Andersen, 2011).

Currently, there are various snow cover datasets for the Asian Water Tower region, such as the Interactive Multi-sensor Snow and Ice Mapping System (IMS) (Mazari et al., 2013), MODIS/Terra Snow Cover Daily L3 Global 500 m SIN Grid

product (MOD10A1) (Hall and Riggs, 2016), MODIS Snow-Covered Area and Grain size product (MODSCAG) (Painter et al., 2009), Japan Aerospace Exploration Agency (JAXA) long-term snow cover extent dataset (JASMES) (Hori et al., 2017), Northwest Institute of Eco-Environment and Resources (NIEER), Chinese Academy of Sciences AVHRR/MODIS snow cover extent product (NIEER AVHRR/MODIS SCE) (Hao et al., 2021, 2022), Tibetan Plateau long-term daily gap-free snow cover product based on the Hidden Markov Random Field model (HMRFS-TP) (Huang et al., 2022), and the European Space Agency

(ESA) Snow Climate Change Initiative (Snow CCI: MODIS (Nagler et al., 2022)  and AVHRR (Naegeli et al., 2022)). The IMS, JASMES, and NIEER AVHRR SCE products are binary products and have relatively coarse resolutions. The NIEER AVHRR/MODIS SCE and HMRFS-TP products are binary, and their range cannot fully cover the Asian Water Tower area. The MODSCAG product has better accuracy, but it cannot freely available. The MOD10A1 product is the most widely used, but it has data gaps, and the linear relationship between the NDSI and FSC is not always valid. Compared to MODSCAG and

MOD10A1, the AWT MODIS FSC product has overall better accuracy in the AWT region(Hao et al., 2019). Although Snow CCI is a fractional snow cover product, the key parameter for retrieving canopy transmittance is calculated using static forest data from early the 2000s, which makes it difficult to capture the dynamic changes in snow cover in forest areas, and it has data gaps. Therefore, there is an urgent need for a high precision, high spatiotemporal resolution, and long-term series cloud-gap-filled fractional snow cover dataset to meet the growing demand for snow monitoring in the Asian Water Tower region.

This study used the MESMA-AGE algorithm and the MSTI algorithm to produce a MODIS long-term series daily fractional snow cover dataset for the Asian Water Tower region from 2000 to 2022. This work is organized as follows: first, the study area and datasets are presented. Then, the MESMA-AGE algorithm framework, the MSTI algorithm framework, and the data processing process are introduced. The two algorithms are used to produce a daily cloud-gap-filled fractional snow cover dataset for the Asian Water Tower region. Finally, the accuracy of this product is evaluated using high spatial resolution

Landsat-8 images and meteorological station snow depth data from the China Meteorological Administration (CMA).





## 2 Study area and data

### 2.1 Study area

The Asian Water Tower region consists mainly of the Pamir Plateau, Xinjiang, and the Qinghai-Tibet Plateau in China (Fig.1). The latitude and longitude ranges are 24° - 54° N and 60° - 106° E respectively, with an average elevation of over 4000 meters. The Asian Water Tower region is the birthplace of more than 10 major rivers in Asia, sustaining nearly 2 billion people in its vicinity(Immerzeel et al., 2020; Li et al., 2022). In the Asian Water Tower region, the monthly average snowmelt runoff ratio is greater than 30% in more than half of the months, far exceeding the surrounding area (Yang et al., 2022). Over the past 50 years, the temperature in Asia Water Tower region has increased by an average of 0.3 to 0.4 °C per 10 years, which is twice the global average rate (Barnett et al., 2005; Kraaijenbrink et al., 2017). As one of the most important climate response factors (Liu and Chen, 2000; Immerzeel et al., 2010), the distribution and change in snow cover is of great importance for the study of climatic and ecological changes across the region. Meanwhile, fractional snow cover data are important input for the Snowmelt Runoff Model (SRM) (Martinec, 1975) and can also be used for snow water equivalent reconstruction and optimization (Rittger et al., 2016). Therefore, a set of high-precision fractional snow cover products is necessary for hydrological simulation and hydrological applications in the Asian Water Tower region.

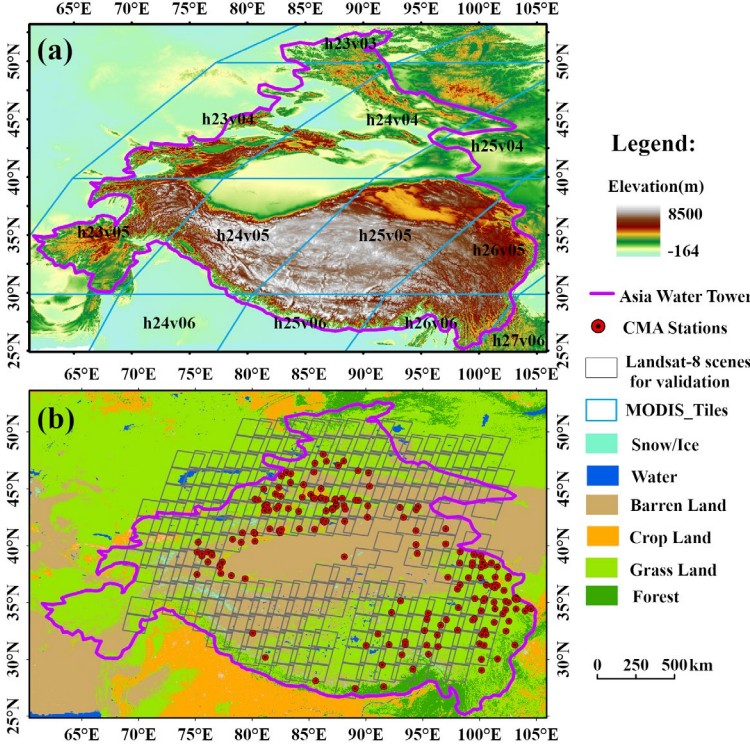

**Figure 1:** DEM (a) and Land cover (b) maps of the AWT with the positions of the MODIS tiles, Landsat scenes and CMA stations used in the validation



## 2.2 MODIS surface reflectance data

This study used MODIS surface reflectance products MOD09GA and MYD09GA in Collection 6 spanning from 2000 to
2022. These surface reflectance products have two data layers. The 500 m reflectance data layer provides reflectance, quality
assessment level, observation area, observation number, and 250 m scan information for bands 1-7. The 1 km geographic
information data layer provides additional information, such as observation times, quality assessment levels, sensor azimuth
zenith angles, solar azimuth altitude angles, and orbit pointers. In addition, it also includes metadata information of the file
(production information, geographical scope, etc.). The characteristics of MODIS solar reflective bands are shown in table 1.
The cloud information used in this study was obtained from the 'state_1km' layer, which includes ' cloud state ' is not a clear
and ' cirrus detected ' is a high. Moreover, 12 MODIS tiles('h23v03', 'h23v04', 'h23v05', 'h24v04', 'h24v05', 'h24v06', 'h25v04',
'h25v05', 'h25v06', 'h26v05', 'h26v06', and 'h27v06')with sinusoidal projection were used in this study, as shown in Figure 1
(a).

**Table 1:** MODIS spectral characteristics

| Band Name | Spectral Range(μm) | Central Wavelength(μm) | Spatial Resolution(m) |
|-----------|--------------------|------------------------|------------------------|
| band1 | 0.62-0.67 | 0.645 | 500 |
| band2 | 0.841-0.876 | 0.858 | 500 |
| band3 | 0.459-0.479 | 0.469 | 500 |
| band4 | 0.545-0.565 | 0.555 | 500 |
| band5 | 1.23-1.25 | 1.24 | 500 |
| band6 | 1.628-1.652 | 1.64 | 500 |
| band7 | 2.105-2.155 | 2.13 | 500 |

## 2.3 Landsat-8 images

This study used the Google Earth Engine (GEE) cloud platform to select a total of 2745 scenes of Landsat-8 images from
2013 to 2021 that met the cloud coverage ratio of less than 10% and snow coverage ratio of more than 30% as "ground truth"
to validate our fractional snow cover product. Landsat-5 Thematic Mapper (TM) has obvious attenuation since 2000, and the
Landsat-7 Enhanced Thematic Mapper Plus (ETM+) sensor has been affected by striping in 25% of the image area due to
135    scanner failure since June 2003. Therefore, this study mainly focused on Landsat-8 images to evaluate the accuracy of the
AWT MODIS FSC dataset. To better evaluate the MESMA-AGE algorithm and the AWT MODIS FSC product, this study
also applied the MESMA-AGE algorithm to retrieve Landsat-8 fractional snow cover, which has been demonstrated to have
good accuracy on Landsat-8 using higher resolution Gaofen-2 imagery with an OA of 94.46% and RMSE of 0.094 (Hao et al.,
2019). The Landsat-8 fractional snow cover results at 30 m were resampled to the resolution of the AWT MODIS FSC product
140    (0.005°) through aggregation and averaging. Subsequently, the Landsat fractional snow cover results at 0.005° resolution were
used to assess the accuracy of the MODIS clear sky retrieval results in the AWT MODIS FSC product.



## 2.4 Ground snow-depth measurements

As Landsat images can only assess the accuracy under clear sky, this study chose to use snow depth data from meteorological stations to support the validation of the accuracy of the spatio-temporal reconstruction results under cloud cover. This study used a total of 175 *in situ* stations provided by the China Meteorological Administration in the Asian Water Tower area from 26 February 2000 to 30 April 2019, as shown in Figure 1b. Figure 1b shows that the *in situ* stations are more evenly distributed in the southeast of the Qinghai Tibet Plateau, Tianshan Mountains and Altay Mountains, where seasonal snow is prevalent. Snow depth data are measured in an open field at 8:00 am using a professional meter ruler. If the fractional snow cover is greater than 50% and the snow depth is greater than 1 cm, it is considered snow and recorded. The geographical coordinates, time of observation, and snow pressure are also recorded. In this study, a threshold of 3 cm was used to classify snow using *in-situ* snow depth data(Huang et al., 2022), where a snow depth less than 3 cm was reclassified as no snow; otherwise, it was recognized as snow. To further illustrate the accuracy of snow identification, this study excluded stations with snow depths greater than 1 cm but snow cover days less than 20 (Zhang et al., 2020; Hao et al., 2021).

## 2.5 Auxiliary data

To better evaluate the accuracy of the MESMA-AGE algorithm and the AWT MODIS FSC product, auxiliary information, such as elevation and the land cover type of the Asian Water Tower, were used. The GEE cloud platform provided the MCD12Q1 V6.1 annual International Geosphere-Biosphere Programme (IGBP) classification data (Sulla-Menashe et al., 2019). The surface types were further divided into four categories: bare land, grassland, forest, and plateau mountain. The GEE cloud platform was utilized to obtain Shuttle Radar Topography Mission (STRM) digital elevation model (DEM) data. The DEM data were then resampled from 90 m to the 0.005° resolution of the AWT MODIS FSC product(Reuter et al., 2007).

## 3 Methodology

Figure 2 shows the flowchart of the AWT MODIS FSC production. According to the accuracy evaluation of the MOD10A1, MODSCAG, and MODAGE fractional snow cover products in the Qinghai Tibet Plateau region, the MODAGE product had the highest accuracy (Hao et al., 2019). Therefore, the MODAGE fractional snow cover retrieval algorithm (MESMA-AGE algorithm) was selected for the fractional snow cover retrieval of Terra and Aqua MODIS surface reflectance version 6 data in the Asian Water Tower region. Second, based on the Terra/MODIS fractional snow cover retrieval results, the Aqua/MODIS fractional snow cover retrieval results were used to fill in data gaps due to clouds and missing observations(Li et al., 2014). Third, the Geospatial Data Abstraction Library (GDAL) was used to reproject and mosaic the fractional snow cover retrieval results of 12 MODIS tiles (GDAL Development Team, 2022). Fourth, the MSTI algorithm was developed for performing spatiotemporal interpolation on pixels with cloud cover or missing data, enabling the generation of a daily cloud-gap-filled fractional snow cover product. Finally, accuracy evaluation and algorithm optimization of the MESMA-AGE algorithm and the AWT MODIS FSC product were performed using snow depth data from meteorological stations and Landsat-8 imagery.



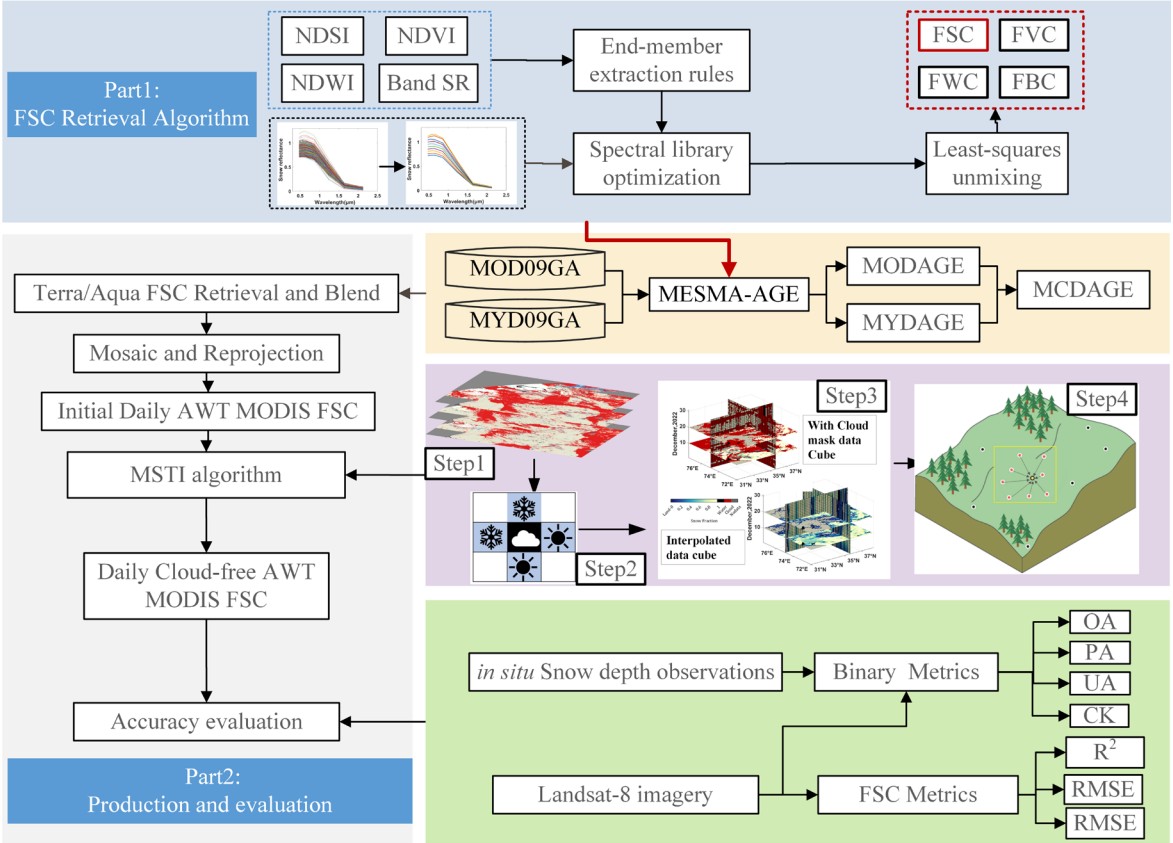

**Figure 2:** Overall flowchart of the AWT MODIS FSC product

## 3.1 MESMA-AGE algorithm

When a pixel contains information from multiple surface types, it is called a mixed pixel, whereas a pixel containing only one type of ground object can be called an endmember of that surface type. The algorithm for unmixing mixed pixels is mainly based on the linear combination of the spectral information of the endmember(Roberts et al., 1998). To analyze the spectral information combination of a pixel, a linear spectral mixing analysis model can be used, which assumes that different endmember energies only undergo single scattering mixing and that there is no nonlinear mixing process(Painter et al., 2003). The linear spectral mixing analysis expression and constraints can be expressed as Equations 1-3:

$$R_\lambda = \sum_{i=1}^{N} F_i R_{i,\lambda} + \varepsilon_\lambda \tag{1}$$

$$\sum_{i=1}^{N} F_i = 1 \tag{2}$$

$$F_i \geq 0 \tag{3}$$





Painter et al. (2003) established an endmember spectral library by collecting spectra of various types of vegetation, rocks, soils, and lake ice from the field and lab and optimized the endmember metadata of snow cover of different grain sizes using radiative transfer models. They then used this spectral library with MODIS images to produce the MODIS Snow Covered Area

185 and Grain Size algorithm (MODSCAG)(Painter et al., 2009). Due to the phenomenon of "the same object with different spectra", spectra from limited observation conditions in the field and lab have difficulty representing the actual complex surface. Meanwhile, the spectrum simulated by the Mie/DISORT model can represent the reflection characteristics of snow under different snow properties and observation conditions, but it is also susceptible to the simulation errors of the model itself. In this study, the MEAMA-AGE algorithm was used to retrieve the fractional snow cover in the Asian Water Tower region,

190 which combines an image-based automatic endmember extraction algorithm(Shi, 2012) with a spectral library optimization method (Xu et al., 2015). The MESMA-AGE algorithm can improve the computational efficiency while ensuring the representativeness of the endmembers(Hao et al., 2019). The rules for extracting snow and non-snow endmembers are shown in Table 2.

**Table 2.** Endmember extraction rule of the MESMA-AGE algorithm

| End-Member | Rule for MODIS Surface Reflectance data |
|---|---|
| Snow | NDSI>0.75 & NDVI<-0.035& $R_{0.55}$>0.7 |
| Vegetation | NDSI<-0.4 & NDVI>0.7 |
| Soil/rock | NDSI<-0.4 & 0<NDVI<0.15 |
| Waterbody | NDWI>0. 2& $R_{0.86}$<0.2 |

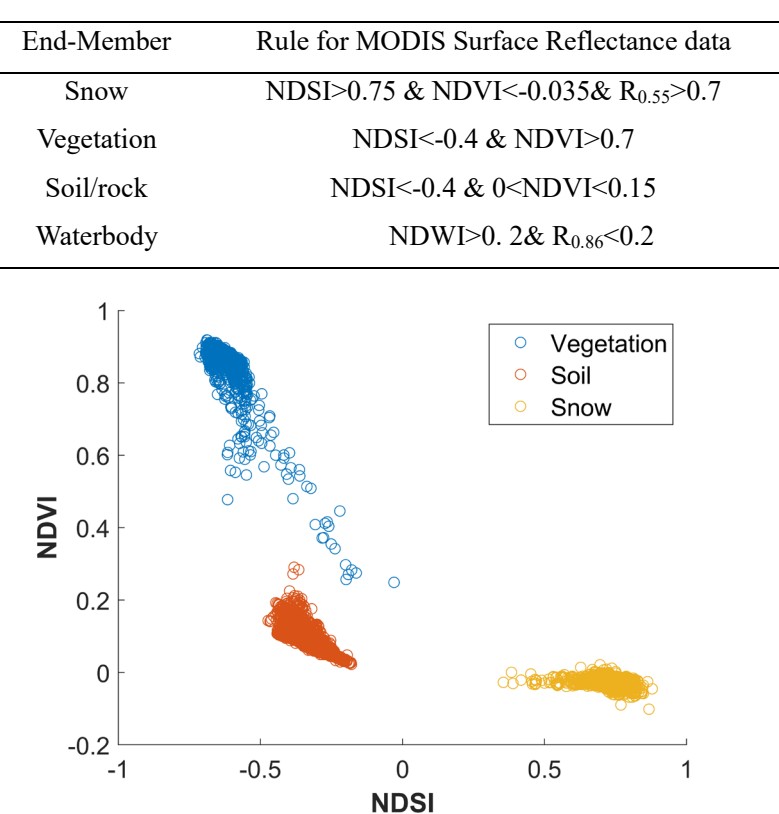

195

**Figure 3:** The NDSI and NDVI pattern of vegetation, soil/rock, and snow endmembers from MODIS images





For this study, 8893 samples were selected from the MODIS data in January, November, and December 2020. The sample types include snow (3136 samples), vegetation (1116 samples), and soil (4641 samples). The samples were used to create the ground object feature map (Fig. 3). The x-axis in Fig. 3 is the NDSI value, and the y-axis is the NDVI. From Figure 3, snow, vegetation, and bare ground pixels can be easily located in the NDSI-NDVI feature space. The endmember extraction rules in Table 2 can effectively achieve endmember extraction for different land types.

### 3.2 Multistep spatiotemporal interpolation algorithm

This study developed a multistep cloud removal algorithm that combines temporal and spatial information. The MSTI algorithm prioritizes the use of nearby spatiotemporal information based on the characteristics of snow cover and achieves complete cloud removal for still cloudy pixels by further expanding their spatiotemporal range. This algorithm is mainly divided into four steps: temporal filtering with a $3\times3$ temporal window, $4\times4$ spatial interpolation, piecewise cubic hermite interpolating polynomial (PCHIP) for the 19-day period, and further spatial interpolation using a $10\times10$ window. The process is shown in the MSTI algorithm flowchart in Figure 2.

**1)** The temporal filtering algorithm assumes that FSC does not change during a short period. (Hou et al., 2019). In previous studies, the size window of the adjacent time filter ranged from 1 d to 8 d. Due to the unique climate conditions and terrain conditions of the Asian Water Tower (high wind speeds can easily redistribute snow, and thin layers of snow can melt and sublime quickly), the snow cover changes rapidly. Therefore, choosing a longer time window may introduce errors. In this study, the time window of the adjacent time filtering algorithm was set to 3 d (the day of cloud cover, and the day before and after cloud cover). If a given pixel is covered by clouds and there are no clouds before and after two days, the FSC value of the cloud pixel can be calculated using the following formula:

$$FSC\_predict^T_{cloud}(x,y) = (FSC_{observed}{}^{T-1}_{cloud-free}(x,y) + FSC_{observed}{}^{T+1}_{cloud-free}(x,y))/2 \qquad (4)$$

where $FSC\_predict^T_{cloud}(x,y)$ is the predicted FSC value of cloud pixels $(x,y)$ at time T, $FSC\_observed^{T-1}_{cloud-free}(x,y)$ is the observed FSC value of cloud-free pixel $(x,y)$ at time T-1, and $FSC\_predict^{T+1}_{cloud}(x,y)$ is the observed FSC value of cloud-free pixel $(x,y)$ at time T+1.

**2)** Based on the continuity of snow cover in spatial continuity, snow cover can be interpolated based on information from non-cloud pixels around a cloud pixel (Gafurov and Bárdossy, 2009; Paudel and Andersen, 2011; Lindsay et al., 2015). Considering the situation where there are at least 3 identical pixels in the 4 pixels adjacent to a cloud pixel, the cloud pixel discrimination rule is as follows: if at least 3 of the 4 pixels above, below, left, and right of a cloud pixel are covered with snow, the central cloud pixel is assigned the mean FSC of the snow pixels in the adjacent 8 pixels. If at least 3 out of the 4 pixels above, below, left, and right of a cloud pixel are land, the central cloud pixel is assigned as land. In other cases, the pixels retain their cloud pixel values.

**3)** For the remaining cloud pixels after the previous two steps, the PCHIP algorithm is used to interpolate the time series of the missing data. Compared with the spline curve used in previous studies (Dozier et al., 2008; Tang et al., 2013), the PCHIP



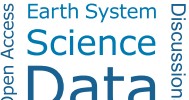

algorithm ensures the monotonicity of the interpolation result, which allows it to achieve spatiotemporally continuous fractional snow cover while suppressing the effects of noise. Through statistical analysis of cloud cover in the Asian Water

Tower region over the past approximately 20 years, 78.92% of the regions has less than 16 days of cloud cover, and 93.14% has less than 19 days of cloud cover. Therefore, the time window selected for this study was set to 9 days before and after the presence of cloud cover pixels.

The PCHIP algorithm assumes that the known function $f(x)$ satisfies $f(x_i) = f_i$ and $f'(x_i) = f_i'$ $(i = 0,1,2, \dots , n)$ at n+1 distinct nodes $x_i$ $(i = 0 \dots , n\ n)$ on the interpolation interval [a, b]. A segmented cubic Hermite interpolation function $G(x)$

can be constructed to satisfy Equations (5), (6) and (7).

$$\text{The polynomial degree of } G(x) \text{ between each cell is 3} \tag{5}$$

$$G(x) \in C^1[a, b] \tag{6}$$

$$G(x_i) = f(x_i), G'(x_i) = f'(x_i), i = (0,1, \dots , n) \tag{7}$$

The expression of $G(x)$ between cells $[x_k, x_{k+1}]$ can be directly obtained from the above conditions:

$$
\begin{aligned}
G(x) = {} & \left(1 + 2\frac{x - x_k}{x_{k+1} - x_k}\right)\left(\frac{x - x_{k+1}}{x_k - x_{k+1}}\right)^2 y_k + \left(1 + 2\frac{x - x_{k+1}}{x_k - x_{k+1}}\right)\left(\frac{x - x_k}{x_{k+1} - x_k}\right)^2 y_{k+1} \\
& + (x - x_k)\left(\frac{x - x_{k+1}}{x_k - x_{k+1}}\right)^2 y_k' + (x - x_{k+1})\left(\frac{x - x_k}{x_{k+1} - x_k}\right)^2 y_{k+1}'
\end{aligned}
\tag{8}
$$

where $x_k$ and $x_{k+1}$ are the positions of two adjacent time points to be interpolated,$y_k$ and $y_{k+1}$ are the FSC corresponding to the two observations before and after the corresponding interpolation point, and $y_k'$ and $y_{k+1}'$ are the corresponding

derivatives.

**4)** After the first three steps of spatiotemporal interpolation, there are still a few cloud pixels left. In this study, the observation information from the 11*11 interpolation window centred on the cloud pixel was used based on the inverse distance weight (IDM) interpolation algorithm, which considers elevation information for spatial interpolation. The IDW interpolation algorithm is an important application of the first law of geography, which uses the distance between the interpolation point

and the sample point as the weight for weighted averaging (Zhao et al., 2022). The closer the interpolation point is, the greater the weight assigned to the sample point. According to existing studies, elevation is important for the distribution of fractional snow cover (Li et al., 2017), but traditional IDW algorithms only consider spatial distance. Therefore, this study incorporated the influence of elevation on fractional snow cover on this basis. The SNOWL method is a commonly used algorithm for spatiotemporal interpolation of snow cover, and scholars often use 100 m as the interval(Huang et al., 2016; Li et al., 2017).

Therefore, this study mainly used clear sky pixel information within the range of elevation differences less than 100 m around the pixels. The process of the IDW interpolation algorithm considering elevation information is described in Eq. (9), Eq. (10) and Eq. (11):

$$d_i = \sqrt[2]{(x - x_i)^2 + (y - y_i)^2} \tag{9}$$



$$w_i = \frac{\Delta E_i / d_i}{\sum_1^n \Delta E_i / d_i}, \Delta E_i = \begin{cases} 1 - \frac{|\Delta Elev_i|}{100}, |\Delta Elev_i| \leq 100 \ m \\ 0, |\Delta Elev_i| > 100 \ m \end{cases} \tag{10}$$

$$FSC(x, y) = \sum_{i=1}^n w_i * FSC(x_i, y_i) \tag{11}$$

where $(x, y)$ is the position of the cloud pixel, $(x_i, y_i)$ is the observing pixel positions for the surrounding clear sky pixels, $d_i$ is the distance between the sample point and the position to be interpolated, $\Delta E_i$ is the weight of the i-th sample point obtained

based on elevation, $\Delta Elev_i$ is the elevation difference, $w_i$ is the weight of the i-th sample point, $FSC(x_i, y_i)$ is the fractional snow cover for clear sky pixels, and $FSC(x, y)$ is the interpolated fractional snow cover for the cloud pixel.

**3.3 Evaluation Metrics**

Selected metrics for validation of the AWT MODIS FSC product included the OA, PA, UA, CK, $R^2$, RMSE, and MAE, which are defined below:

$$OA = \frac{TP + TN}{TP + TN + FP + FN} \tag{12}$$

$$PA = \frac{TP}{TP + FP} \tag{13}$$

$$UA = \frac{TP}{TP + FN} \tag{14}$$

$$CK = \frac{OA - P}{1 - P} \quad , P = \frac{(TP+FN)(TP+FP)+(TN+FN)(TN+FP)}{(TP+TN+FP+FN)^2} \tag{15}$$

$$R^2 = \frac{\left[\sum(f_{est} - \overline{f_{est}})(f_{ref} - \overline{f_{ref}})\right]^2}{\sum(f_{est} - \overline{f_{ref}})^2 \cdot \sum(f_{ref} - \overline{f_{ref}})^2} \tag{16}$$

$$MAE = \frac{1}{N} \sum_N |f_{est} - f_{ref}| \tag{17}$$

$$RMSE = \sqrt{\frac{1}{N} \sum_N (f_{est} - f_{ref})^2} \tag{18}$$

where TP indicates true positive, TN indicates true negative, FP indicates false positive, FN indicates false negative, $f_{est}$ indicates fractional snow cover estimation derived from MODIS, and similarly, $f_{ref}$ indicates reference fractional snow cover derived from Landsat–8. To calculate the overall accuracy, binary snow cover was labeled for pixels with fractional snow cover ≥15%(Rittger et al., 2013; Wang et al., 2019).





## 4 Results

In this study, a comprehensive evaluation of the accuracy of the AWT MODIS FSC product in two dimensions, namely binary and fractional snow cover, was conducted. The binary and fractional snow cover accuracies of the AWT MODIS FSC product were quantitatively assessed under three different conditions: overall, different surface types, and altitudes. This evaluation was performed using 2745 Landsat-8 images. Additionally, the binary accuracies of the AWT MODIS FSC product were specifically evaluated in three different conditions, i.e., overall (including clear and cloud conditions), clear sky, and

cloud cover, utilizing snow depth data from 175 meteorological stations.

### 4.1 Validation with Landsat-8 images

### 4.1.1 Overall results

    In this study, the 2745 Landsat-8 scenes were used as "ground truth" to quantitatively evaluate the AWT MODIS FSC product obtained from clear sky in two dimensions: binary (OA, PA, and UA) and fractional snow cover ($R^2$, MAE, and

RMSE). Figure 4 shows violin charts of the accuracy evaluation metrics, and the violin charts of each accuracy metric are composed of two parts: the outer violin chart and the inner box plot. The left side of the outer violin chart is the kernel density map, the larger the area of a certain range is; the greater the probability of the distribution near a certain value, and the horizontal line in the left area is where the median is located. To the right of the violin plot is a histogram of the frequency of a value, i.e., the longer the line is, the more points there are for that value. The internal box plot contains a gray rectangle

consisting of the upper and lower quartiles, with the mean position represented by the white point. From Figure 4, the minimum OA value of the AWT MODIS FSC product is 80.38%, and the average value is 95.17%. The minimum PA value is 58.71%, and the average value is 97.34%, i.e., the average omission error of this product is 2.66%. The minimum UA value is 67.02%, and the average value is 97.59%, i.e., the average commission error is 2.41%. The $R^2$ value distribution range is 0.40-0.97, and the average value is 0.80. The MAE ranges from 0.01 to 0.23, with an average of 0.10. The RMSE ranges from 0.02 to 0.26,

with an average of 0.16. The above results provide a good illustration of the consistency of the AWT MODIS FSC product with the "ground truth" data.

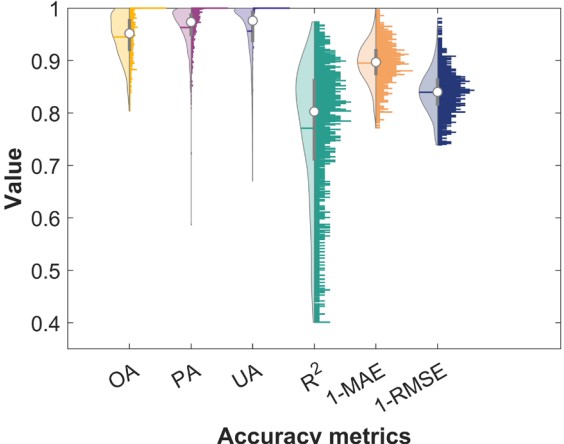

**Figure 4:** Violin charts for the accuracy evaluation metrics of the AWT MODIS FSC product validated by Landsat images

To better explore the interannual differences in each accuracy metric, this study divided the 2745 Landsat-8 scenes by year.
Figure 5 shows the interannual distribution of each accuracy metric, and Table 3 shows the number of Landsat images and the interannual average of each accuracy metric. Figure 5 and Table 3 show that the interannual means of the OA range from 92.41% to 96.22%, PA from 94.77% to 97.88%, UA from 95.35% to 98.55%, $R^2$ from 0.76 to 0.81, MAE from 0.09 to 0.11, and RMSE from 0.15 to 0.17. The accuracy metrics perform better except for 2013, where the poor accuracy indicators are mainly due to the overall low number of validation data and the fact that a significantly larger proportion of the validation data
are located at high altitudes in mountainous areas than in other years. The results of this part of the retrieval are strongly influenced by topographical effects.

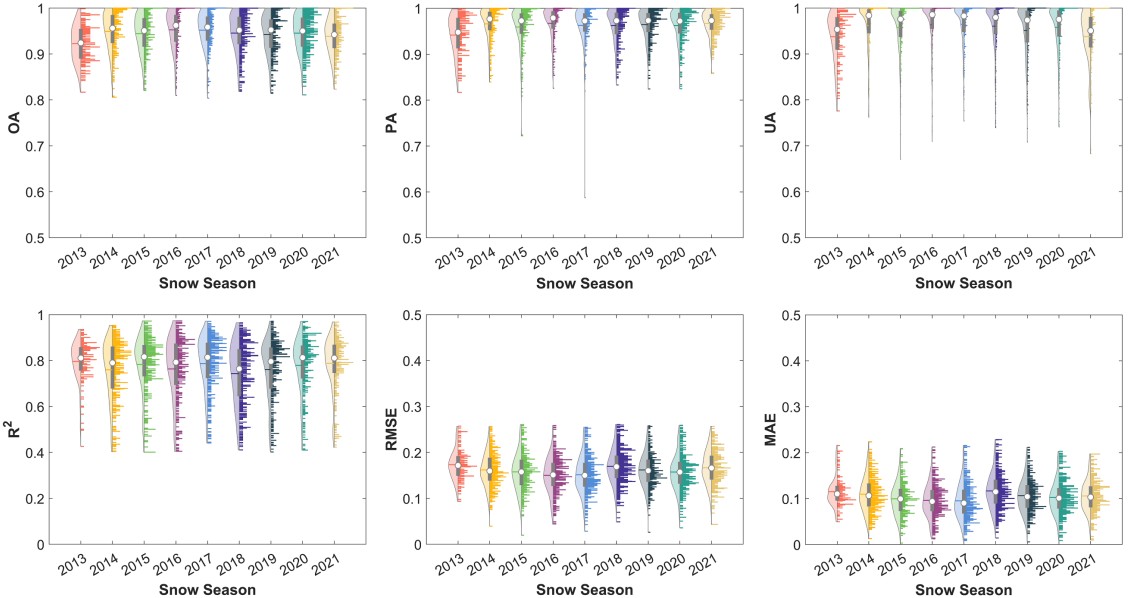

**Figure 5:** Violin chart for the interannual accuracy evaluation metrics





**Table 3.** The number of Landsat images and the interannual average of each accuracy metric

| Year | Image Number | OA | PA | UA | $R^2$ | MAE | RMSE |
|------|------|------|------|------|------|------|------|
| 2013 | 123 | 92.41% | 94.77% | 95.35% | **0.81** | **0.11** | **0.17** |
| 2014 | 328 | 95.60% | 97.67% | 98.38% | **0.79** | **0.11** | **0.16** |
| 2015 | 287 | 95.08% | 97.26% | 97.58% | **0.79** | **0.10** | **0.16** |
| 2016 | 309 | 96.22% | 97.88% | 98.55% | **0.81** | **0.09** | **0.15** |
| 2017 | 315 | 95.85% | 97.25% | 98.27% | **0.79** | **0.09** | **0.15** |
| 2018 | 389 | 95.16% | 97.30% | 97.95% | **0.76** | **0.11** | **0.17** |
| 2019 | 360 | 95.23% | 97.32% | 97.37% | **0.79** | **0.10** | **0.16** |
| 2020 | 320 | 95.01% | 97.21% | 97.53% | **0.81** | **0.10** | **0.16** |
| 2021 | 314 | 94.20% | 97.33% | 95.07% | **0.81** | **0.10** | **0.17** |

### 4.1.2 Evaluation for different land surface types

Spectra of snow-covered surface also encounters the impacts of land cover type. In particular, in forested areas, snow below the forest canopy is difficult to observe with spaceborne sensors because the forest blocks the visible, near-infrared, and shortwave infrared bands(Wang et al., 2021, 2023). Therefore, in this study, 2745 scenes of Landsat-8 images were divided into four categories according to land surface types, namely, grassland area (1500 scenes), bare land area (264 scenes), forest area (410 scenes), and mountain area of the Himalayas and Pamir Plateau (571 scenes). The accuracy evaluation metrics under the two dimensions of binary value and fractional snow cover of the four land surface types are shown in Figure 6. Regarding the accuracy metrics of binarization, the accuracy of the grassland area is better than that of other land surface types, and its average OA is 95.99%, while the bare land area is the worst but also reaches 93.69%. This is because the bare land area is mainly the hinterland of the Asian Water Tower region, which is mainly desert/Gobi with little vegetation growth, and there are also many large and small lakes distributed in this area. The bright land and the winter water surface lead to deviations in the retrieval algorithm. In addition, the snow in this region is relatively broken, and the observation scales of MODIS and Landsat are quite different, making it difficult for MODIS to capture the broken snow information as effectively as Landsat. These two reasons lead to many errors in the results. The UA also illustrates this problem well. Figure 6 shows that the average UA of bare land is only 94.06%, approximately 4% lower than that of the other surface types. From the perspective of fractional snow cover accuracy metrics, grassland and forest are slightly worse, mainly because it is difficult to observe the snow signal shielded by the vegetation canopy at the MODIS scale.



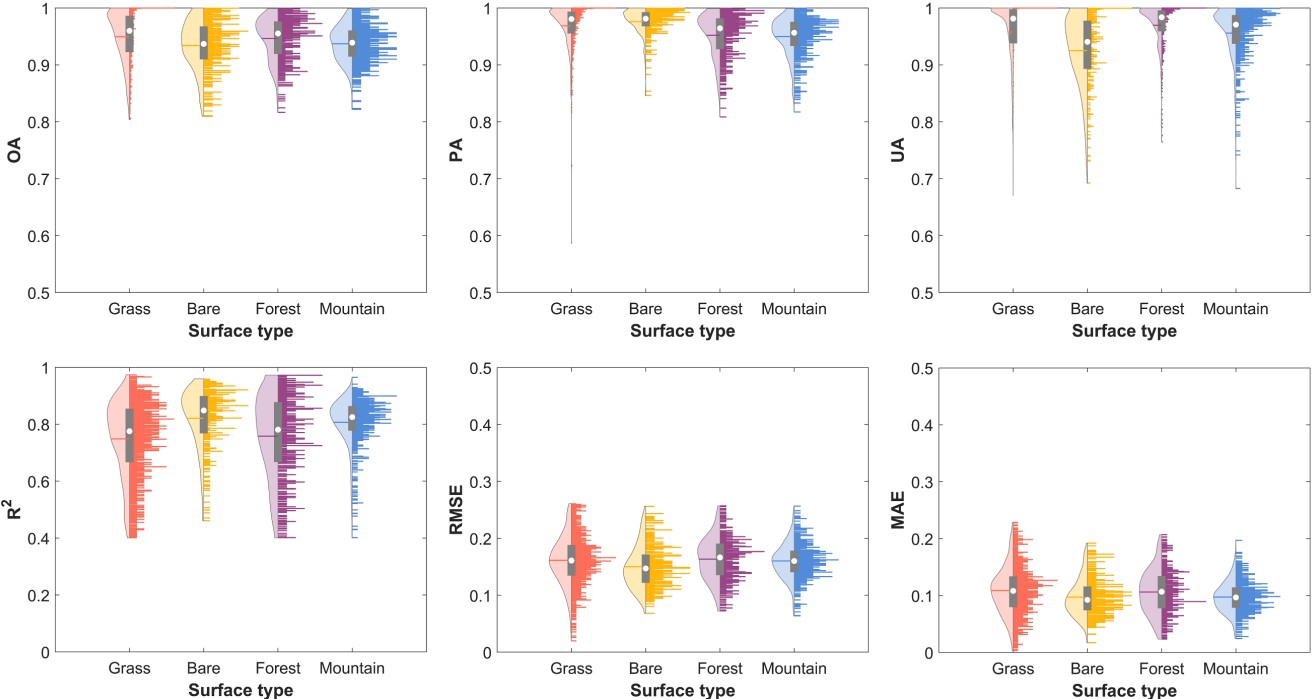

**Figure 6:** Violin charts for accuracy evaluation metrics of the AWT MODIS FSC product validated by Landsat images under different surface types

### 4.1.3 Evaluation for different altitudes

Topographic effects challenge accurate snow cover mapping with optical imagery as well. The snow cover products such as MOD10A1 have been reported uncertaines related with altitudes(Zhang et al., 2020; Wu et al., 2021; Huang et al., 2022). Therefore, the 2745 Landsat-8 images are divided into four sections according to altitude, namely < 3 km (1603 scenes), 3-4 km (395 scenes), 4-5 km (355 scenes) and > 5 km (392 scenes). The results of the accuracy evaluation metrics according to different heights are shown in Figure 7. As shown in Figure 7, both binary and fractional snow cover accuracy metrics show a decreasing trend with increasing altitude. The areas smaller than 3 km are mostly distributed in northern Xinjiang, China, i.e., the area north of 40°N, where the snow distribution is relatively concentrated, and the surface type is mostly grassland with a small amount of forest, so the accuracy is highest. The three elevation regions greater than 3 km are mainly distributed in the Tianshan Mountains, the Pamir Plateau and the Tibetan Plateau. Snow fragmentation and topographic heterogeneity in these regions increase with altitude. This results in a slight reduction in the accuracy of the AWT MODIS FSC product.

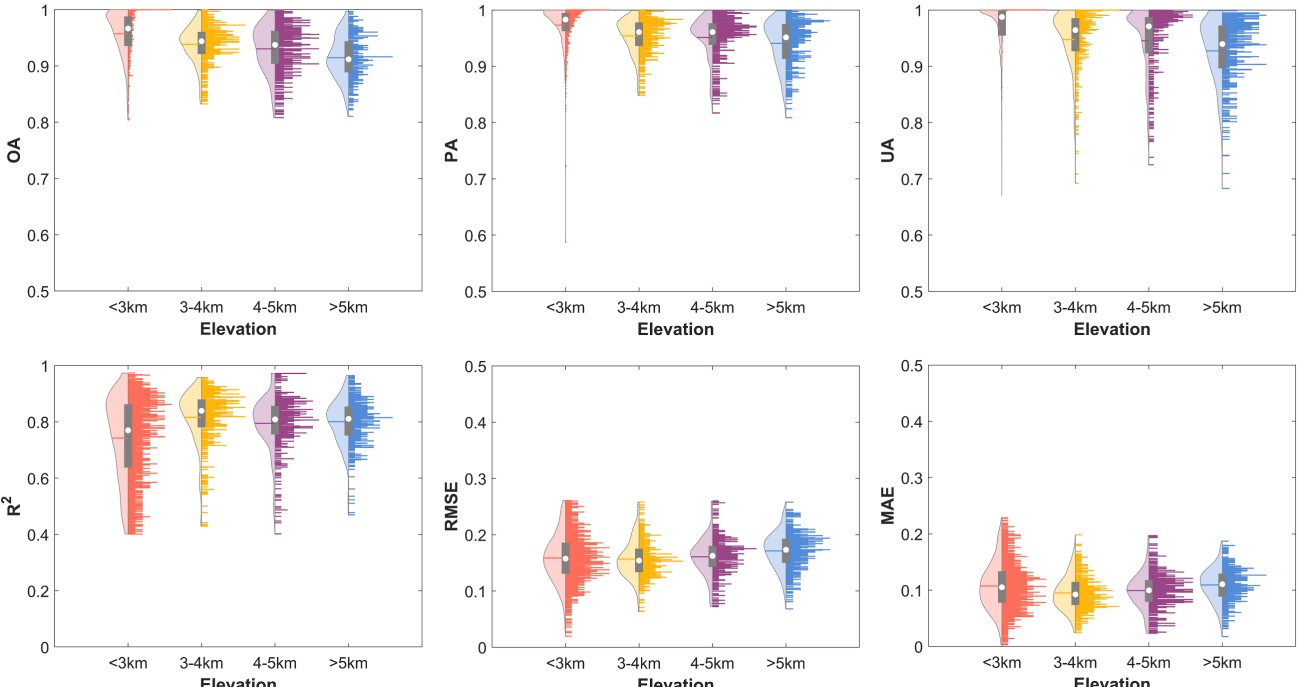

**Figure 7**: Violin chart for accuracy evaluation metrics of the AWT MODIS FSC product validated by Landsat under different altitudes

## 4.2 Validation with *in situ* snow depth measurements

Landsat images can only be used to evaluate the accuracy of the fractional snow cover retrieval algorithm and FSC product

under clear sky conditions, and the fractional snow cover information reconstructed by the MSTI algorithm needs to be verified by snow depth observations at meteorological stations. Therefore, this study used a total of nearly 1 million observations collected from 175 *in situ* stations during the period from 26 February 2000 to 30 April 2019 to evaluate the accuracy under different weather conditions. The numbers of meteorological stations and SD observations obtained each year are shown in Figure 8. The number of stations fluctuates to some extent each year, and the number of snow depth observations available

after 2014 has nearly tripled compared with before 2014.

Earth System
Science
Data

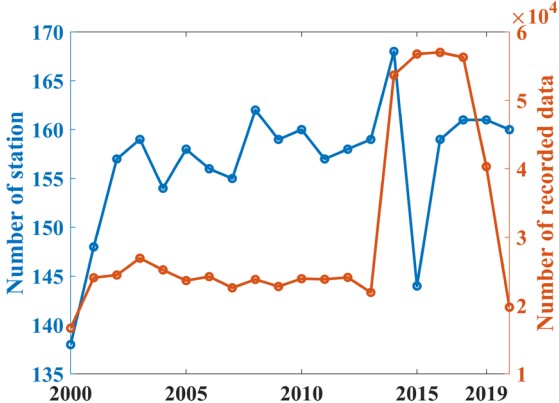

**Figure 8:** The number of *in situ* stations and observed data volume per year (3.12, 2000-4.30, 2019)

### 4.2.1 Overall results

To better assess the accuracy of snow identification, only *in situ* stations with more than 20 days of snow observations per
year were selected for evaluation(Zhang et al., 2020; Hao et al., 2021). This reduced the number of *in situ* stations and the total
observation data by approximately half. In this study, snow depth observations from 175 *in situ* stations were used to perform
a binary evaluation of the AWT MODIS FSC product. Table 4 shows the results of the overall accuracy evaluation. It can be
seen from the results that the OA of the product reaches 93.26%, PA can reach 84.41%, and UA can reach 82.14%, i.e., the
omission error is 15.59%, and the commission error is 17.86%. In addition, CK reach 0.79. The results of the above accuracy
metrics exclude the stations without snow observations, which indicates that the AWT MODIS FSC product has good accuracy
and good consistency with the snow depth observation data of meteorological stations.

**Table 4.** Confusion matrix and accuracy results of the AWT MODIS FSC product based on snow depth measurements from CMA. OA, PA, UA and CK

| Class | | AWT MODIS FSC | |
|---|---|---|---|
| | | **Snow** | **Non-Snow** |
| ***In situ* snow depth measurements** | **Snow** | 102617 | 18946 |
| | **Non-Snow** | 22316 | 468105 |
| **OA** | | 93.26% | |
| **PA** | | 84.41% | |
| **UA** | | 82.14% | |
| **CK** | | 0.79 | |

To verify the stability of the product accuracy over time, this study performed a binary accuracy assessment of the snow
depth observations at each station by year. The overall results of each accuracy metric over the last 20 years are shown in
Figure 9. Each accuracy metric is relatively stable before and after 2014, but there is a large fluctuation in 2014. The OA metric
exhibits the most significant temporal variation. Before 2014, the fluctuation range of OA is 88.69%-92.96%, and after 2014,

the fluctuation range of OA is 95.05-97.54%. Meanwhile, CK and PA increase significantly after 2014. This also indicates that the consistency between the AWT MODIS FSC product and the snow depth observations from meteorological stations
has improved significantly since 2014. The fluctuation in the above accuracy indicators is mainly due to the significant increase in the number of meteorological station observations used in this study after 2014 and the improvement in the accuracy of snow identification, which ultimately leads to a significant increase in OA. The percentage of cloud cover in different years is also shown in Figure 9 below. Combined with Figures 8 and 9, the increase in the amount of station data and observations acquired in this study, coupled with the decrease in cloud cover in the study area, are two factors that lead to better results in
the station-based accuracy assessment after 2014 than before.

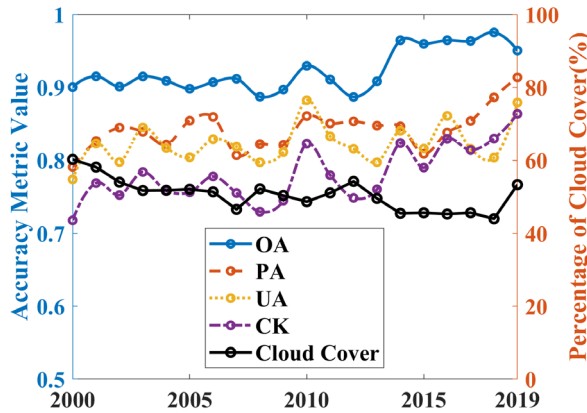

**Figure 9:** Accuracy fluctuations of the AWT MODIS FSC product based on *in situ* snow-depth measurements and the percentage of cloud cover in the past 20 years

### 4.2.2 Accuracy metrics at each *in situ* station

Figure 10 shows the detailed results of the accuracy metrics of the AWT MODIS FSC product verified by the snow depth data of the stations. As shown in Figure 10, the OA of most *in situ* stations is above 90%, with only one *in situ* station below 70%. However, the figure shows that, unlike the OA, the accuracy of the entire Asia Water Tower region is relatively consistent. PA, UA and CK are severely affected by the region. The PA and UA metrics at stations in northern Xinjiang, China, are generally greater than 90%, and CK is also greater than 0.8. This is mainly due to the stable snow cover in the region. The
spatiotemporal reconstruction algorithm of snow cover developed in this research can well grasp the spatiotemporal variation characteristics of snow cover in this region, so that high-precision spatiotemporal reconstruction of snow cover information can be achieved. However, the snow cover in the eastern part of the Asia Water Tower region and the northwestern edge of the Tarim Basin is relatively broken, and the MODIS resolution is coarse. These areas are seriously affected by clouds, so the PA, UA and CK metrics in these areas are generally not high.

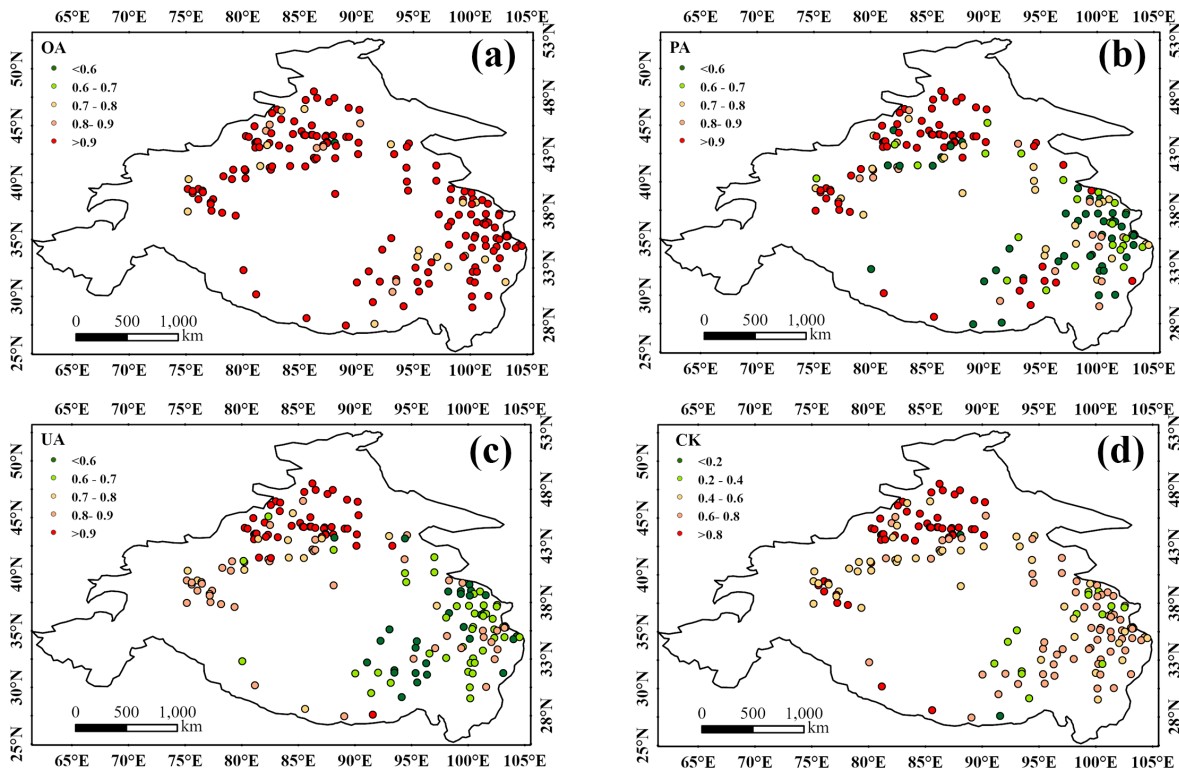

**Figure 10:** Point-based accuracy results of the AWT MODIS FSC product: (a) OA; (b) PA; (c) UA; (d) CK.

### 4.2.3 Performance of spatiotemporal reconstruction algorithm

The above two sections presented the overall accuracy of the AWT MODIS FSC product using snow depth data from meteorological stations. The AWT MODIS FSC product is derived from the composition of two parts: the real MODIS observation under clear sky and the spatiotemporal reconstruction with the MSTI algorithm for cloudy conditions. To further explore the accuracy of the fractional snow cover results of these two parts, the snow depth observation data of the meteorological stations are divided into two categories based on MODIS clear sky and cloudy conditions. First, the stability of the accuracy evaluation metric under clear sky and cloudy conditions is evaluated, respectively and the results are shown in Figure 11. Comparison of Figure 10 shows that there is an increase in accuracy in years with lower cloud cover. Figure 11 (a) presents the interannual variation results of the accuracy metrics of fractional snow cover obtained from MODIS clear sky observations using snow depth observations at the selected meteorological stations. OA, PA, UA and CK all exhibit good stability, and the range of variation in OA is 92.80-99.01%. The range of UA is 82.39-90.26%. After 2014, the two indices of PA and CK improved, with the maximum of PA reaching 95.46% and CK reaching 0.91. Figure 11 (b) shows the interannual variation results of the accuracy metrics of the fractional snow cover result obtained by the spatiotemporal reconstruction of the MSTI algorithm using the snow depth observations at the selected meteorological stations. All accuracy metrics decrease
to some extent compared to the clear sky condition. Influenced by the amount of *in situ* station data used, the spatiotemporal reconstruction results show a relatively obvious jump in 2014. The variation range of OA before 2014 is 84.51-90.00%, but the variation range of OA after 2014 is 92.76-95.60%. PA and CK have interannual variations in the years before 2014, but the value of the years after 2014 has a large increase, with the maximum PA reaching 89.61% and CK reaching 0.83. UA has

only a large interannual variation, and there is no significant jump in 2014. The results show that the accuracy of the fractional snow cover based on clear sky observations is significantly better than that of the spatiotemporal reconstruction. This is mainly due to the presence of clouds over a long period of time and over a large area in most of the Asian Water Tower region, the interpolation of this part of the area relies heavily on the last two steps of the MSTI algorithm. The last two steps of the MSTI algorithm require a larger space-time window to complete the interpolation. However, a larger space-time window introduces

more error, especially for snow cover, which has strong spatial heterogeneity and changes rapidly over time.

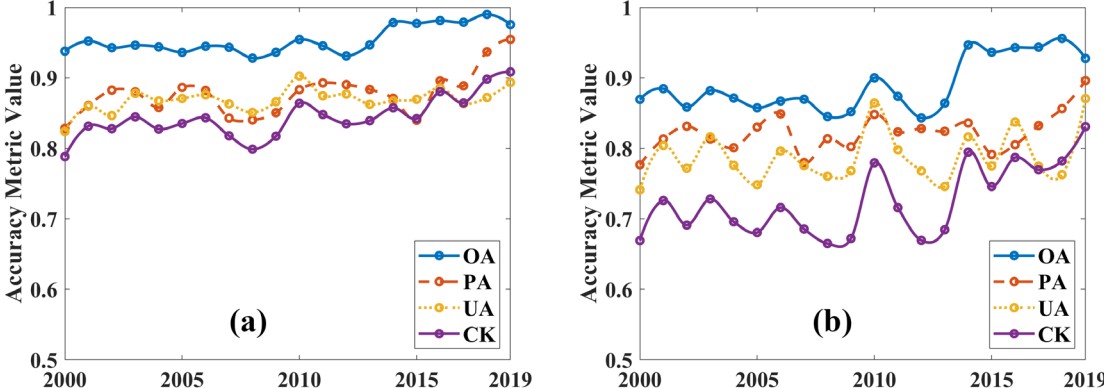

**Figure 11:** Point-based accuracy results of the AWT MODIS FSC product: (a) clear sky observations; (b) spatiotemporal reconstruction.

This study further analyzed the accuracy of the AWT MODIS FSC product obtained from MODIS clear sky and cloud cover

observations at each station, and the results of the binarization accuracy metrics are shown in Figure 12. It can be seen from the figure that the accuracy of fractional snow cover obtained by the clear sky retrievals is significantly better than that obtained by the MSTI algorithm. According to the OA in Figure 12 (a) and (e), the accuracy of the verification results of fractional snow cover based on MODIS clear sky observations is good at all stations, and only a few stations are less than 90%. However, the OA of the fractional snow cover reconstructed by the MSTI algorithm shows some regional differences. When comparing

the accuracy metrics (PA, UA, and CK) of fractional snow cover between the real MODIS observations and the spatiotemporal reconstruction results achieved through the MSTI algorithm, notable regional variations are observed in all three metrics. In other words, the accuracy of the stable snow cover area in northern Xinjiang, China is obviously better than that in the central and eastern parts of the Asia Water Tower region and the northwestern edge of the Tarim Basin, where the snow cover is relatively fragmented and rapidly changing.

**Figure 12:** Point-based accuracy results of the AWT MODIS FSC product under clear sky conditions ((a)OA, (b)PA, (c)UA, and (d)CK) and spatiotemporal reconstruction ((e)OA, (f)PA, (g)UA, and (h)CK)





## 5 Discussion

### 5.1 Comparing AWT MODIS FSC with MOD10A1 and HMRFS-TP product

In order to evaluate the accuracy of AWT MODIS FSC products more objectively, MOD10A1(Hall and Riggs, 2016) and HMRFS-TP(Huang et al., 2022) products are selected as benchmarks for this study using Landsat-8 imagery. MOD10A1 is the most widely used MODIS snow product, which has a long time series (since 2000) and highly spatial and temporal resolution (i.e., 500 m and daily). This study is based on the GEE platform to obtain the MOD10A1 (Collection 61) data from 2013 to 2022 for the Asian water tower region, and uses this data product to evaluate the accuracy difference with the AWT

MODIS FSC product under the clear sky scenario. The HMRFS-TP product is a continuous spatio-temporal binary snow product based on the MOD10A1 product and the HMRFS spatio-temporal interpolation algorithm covering the Tibetan Plateau region within China, and is used in this study to compare the accuracy of the two sets of continuous spatio-temporal products.

The accuracy of the MOD10A1 product depends on the threshold value of NDSI, and the commonly used thresholds are 0.1 (Zhang et al., 2019), 0.29 (Zhang et al., 2021; Tang et al., 2022) and 0.4 (Riggs et al., 2017). Since no spatio-temporal

interpolation is performed for the MOD10A1 product, in order to make a comprehensive and objective comparative assessment, this study uses 1805 Landsat images to compare the clear sky pixels of the two products, and the results are shown in Figure 13 below. The OA, PA and UA of the AWT MODIS FSC product are 97.69%, 98.73% and 98.83% respectively. The NDSI threshold is 0.1, the OA, PA and UA of the MOD10A1 products are 94.51%, 95.70% and 99.04%. The NDSI threshold is 0.29, the OA, PA and UA of the MOD10A1 products are 94. 75%, 98.49% and 96.52%. The NDSI threshold is 0.4, the OA, PA and

UA of the MOD10A1 products are 91.86%, 98.96% and 90.82%. As the NDSI threshold increases, Figure 13 shows that the PA of the MOD10A1 product gradually increases and the UA gradually decreases. This also means that the percentage of omission error is decreasing and the percentage of commission error is increasing. This is because the larger the NDSI, the higher the probability that the image pixel is snow, and the probability of correctly judging snow increases accordingly, which also leads to a lower probability of correctly judging non-snow, resulting in a decrease in overall accuracy. From the results in

Figure 13, the MOD10A1 product has the best accuracy when the NDSI is 0.29, but its accuracy is still lower than that of the AWT MODIS FSC product.

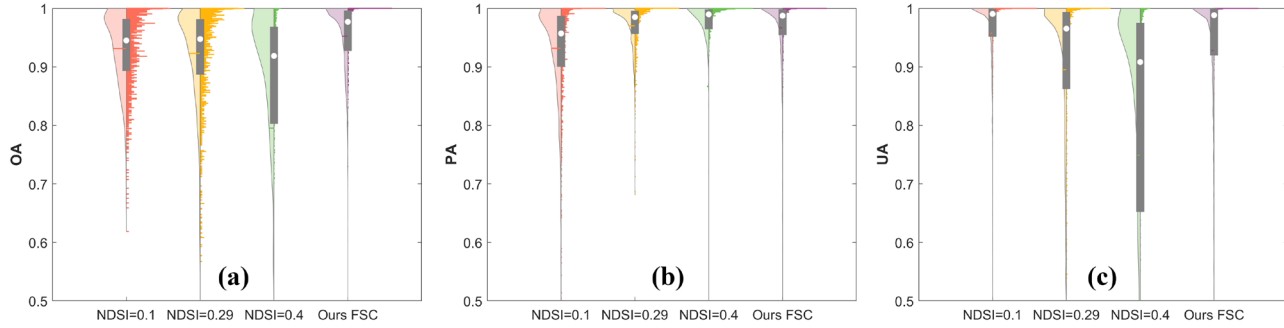

**Figure 13:** Violin charts for the accuracy evaluation metrics of the AWT MODIS FSC and MOD10A1 product validated by Landsat images


The NDSI of the HMRFS-TP product is 0.4 as the threshold for snow identification of MOD10A1, and the spatio-temporal continuous product is obtained by the HMRFS spatio-temporal interpolation algorithm. In this study, 372 Landsat images were used to quantitatively evaluate and compare two sets of spatio-temporal continuous snow products, and the results are shown in Figure 14 below. As shown in Figure 14, the OA, PA and UA of the AWT MODIS FSC product are 89.71%, 94.29% and

86.21%. The OA, PA and UA of the HMRFS-TP product are 79.45%, 99.20% and 66.82%, respectively. Comparing the accuracy indices of the two sets of products, AWT MODIS FSC products are significantly better than HMRFS-TP products, and various accuracy evaluation indices are around 90%. The poor accuracy of the HMRFS-TP products is mainly due to the value of NDSI. Combined with Figure 13, the threshold of 0.4 will lead to serious misclassification of products and is not applicable to the Tibetan Plateau region.

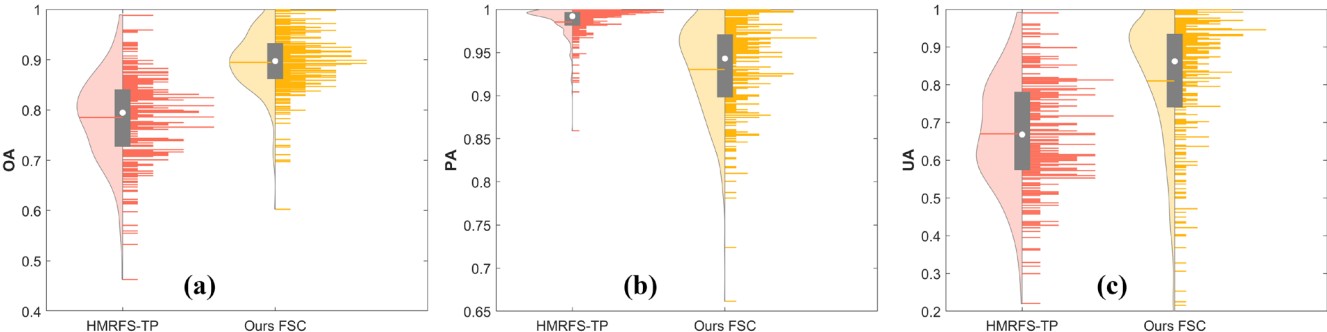


**Figure 14:** Violin charts for the accuracy evaluation metrics of the AWT MODIS FSC and HMRFS-TP product validated by Landsat images

## 5.2 Area differences in fractional snow cover and binary snow cover

Long-term series and high precision fractional snow cover products are of great importance for snow hydrology research in

the Asia Water Tower region. However, most of the existing snow cover products are binary products or space-time gap fill fractional snow cover products. In this study, the clear sky fractional snow cover was retrieved by the MEASMA-AGE algorithm based on MODIS observations. The missing fractional snow cover information caused by cloud cover was reconstructed by the MSTI algorithm, and finally, the spatiotemporally continuous long-term series AWT MODIS FSC product was obtained. The actual snow distribution, which is difficult to capture, was identified with binary values, especially in

scenarios with mixed pixels at medium and coarse resolutions, and the subsequent direct application of binary products will introduce large errors.   Therefore, the AWT MODIS FSC product produced in this study was used to quantitatively analyze the actual difference between the binary snow cover product and the fractional snow cover product, and the results are shown in Figure 15. In this study, pixels with FSC>15% were identified as the binary snow product (Rittger et al., 2013; Wang et al., 2019). Figure 15 (a) shows the difference between the total snow cover area obtained by the binary snow product and that

obtained by the fractional snow cover product in the Asian Water Tower region. There is a significant overestimation of the

binary snow product, with an average difference of 39400 km² and a maximum difference of 102,000 km², which is very large

for the Asian Water Tower region with a total area of only 623,000 km². Figure 15 (b) shows the proportion of the difference

in the total snow area obtained by the two snow cover products in the total snow area obtained by the binary snow product.

The average difference is 34.53%, and the maximum difference is 59.52%. Comparing Figure 15 (a) and (b), the smaller the

total snow cover area is, the greater the difference between the two sets of products, indicating that greater errors in the binary

snow cover product occur for more broken and smaller areas.

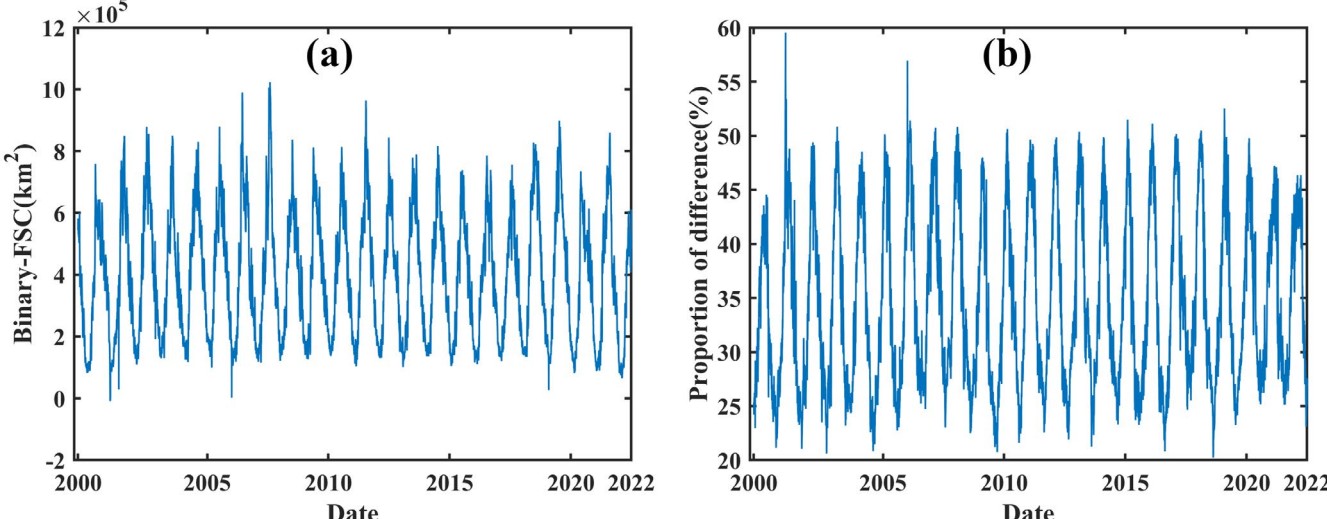

**Figure 15:** Difference between the binary snow cover product and the fractional snow cover product in the Asia Water Tower region: (a)

the difference in total snow cover area between the binary snow cover product and the fractional snow cover product, (b) proportion of

total snow area difference between the two snow products compared with the total snow area obtained by the binary snow product

**5.3 Limitations of the AWT MODIS FSC product**

Clouds in most of the Asia Water Tower region have the characteristics of wide coverage and long duration. If clouds exist

for a long-term, relying only on MODIS data will lead to a serious data gap, and the accuracy of snow cover monitoring will

be reduced, regardless of any spatiotemporal reconstruction algorithm. With the launch of a new generation of geostationary

satellites (FY-4A/B, GOES-17/18, Himawari-8/9, and MSG/MTG), their sensor performance can be comparable to that of

MODIS sensors, and at the same time, the observation can be realized once every 5-15 minutes. Combined with geostationary

sensors, these platforms are expected to provide the highest precision fractional snow cover monitoring. At the same time, the

MODIS cloud product is overestimated, and the observation of multitemporal geostationary satellites can help improve the

MODIS cloud product, thus improving the accuracy of snow cover monitoring. In this study, the monthly average cloud cover

data MODIS and FY4A during the period 2018.04-2022.03 were collected, and the results are shown in Figure 16. Figure 16

(a) shows the average monthly cloud cover statistics from MODIS. It can be seen from the figure that the average monthly

cloud cover in areas with more snow cover, such as the Pamir Plateau, Tianshan Mountain and Altai Mountain, is generally





more than 15 days, and some areas of the Hengduan Mountain range can reach more than 25 days. As shown in Figure 16 (b), the average monthly cloud cover in most regions based on the FY4A data is generally less than 10 days, and the average

monthly cloud cover in the Hengduan Mountain region is also less than 19 days, which will provide strong data support for the subsequent high-precision spatiotemporal reconstruction of fractional snow cover information. However, the large amount of multitemporal observation data from geostationary satellites, the lack of necessary preprocessing steps, such as atmospheric correction, angle correction and geometric registration, and the serious misjudgment of cloud snow in existing cloud products will limit its further application.

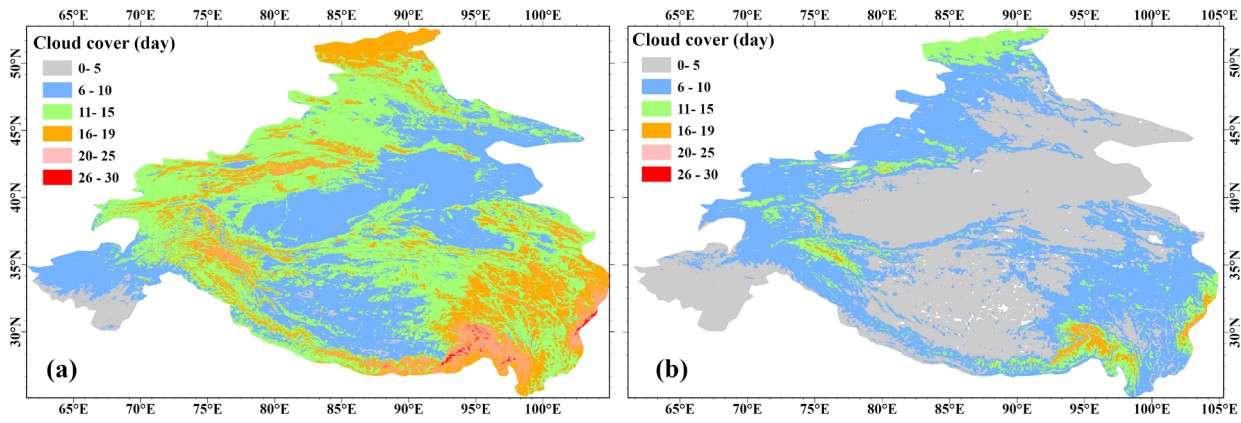


**Figure 16:** Monthly average cloud cover over the Asia Water Tower region (2018.04-2022.03) from (a) MODIS and (b) FY4A

## 6 Data and Code availability

The AWT MODIS FSC product is the daily cloud-gap-filled snow cover data for the Asia Water Tower region. It has a spatial resolution of 0.005° and a daily temporal resolution. This dataset is freely available from the National Tibetan Plateau

Data Center at https://doi.org/10.11888/Cryos.tpdc.272503 (Jiang et al., 2022) or from the Zenodo platform at https://zenodo.org/doi/10.5281/zenodo.10005826. It contains 8347 daily data files from 26 February 2000 to 31 December 2022 in NetCDF format. The filename rule is 'AWT_MODIS_FSC_yyyymmdd.NC', where AWT_MODIS_FSC represents the daily cloud-gap-filled MODIS fractional snow cover product over the Asian Water Tower region, and yyyymmdd indicates the year, month, and day of the data. The dataset contains two layer, 'fSCA' layer: fractional snow cover (non-snow (0), snow

(1-100), water (237), cloud (250), and filling value (255)), 'QA' layer: cloud mask (0: clear sky, 1: cloud mask and 2: invalid value).

The Landsat-8 fractional snow cover dataset for verification is available on the Zenodo platform: https://doi.org/10.5281/zenodo.10008227. The binary value (snow/no-snow) snow depth dataset based on ground stations is available on the Github platform: https://github.com/FangboPan/AWT_Site_SD. The code is available on the Github platform:

https://github.com/FangboPan/AWT_MODIS_DailyFSC_Product_code_v1.



## 7 Conclusions

In this study, based on the MESMA-AGE algorithm and the MSTI spatiotemporal reconstruction algorithm, the daily AWT MODIS FSC product was produced with long-term series, high precision, and spatiotemporal continuity in the Asian Water Tower region. The spatial resolution of the product is 0.005° from 2000 to 2022. The new AWT MODIS FSC product was

quantitatively evaluated in two dimensions: binary value and fractional snow cover using snow depth observations from meteorological stations and high spatial resolution Landsat-8 images. Based on the results of the Landsat-8 image accuracy evaluation, the binarized identification accuracy metrics OA, PA and UA are 95.17%, 97.34% and 97.59%, respectively. The total fractional snow cover accuracy metrics $R^2$, RMSE and MAE are 0.80, 0.16 and 0.10, respectively, compared with 2745 Landsat-8 images. All these results indicate that the AWT MODIS FSC product has good consistency with the high spatial

resolution Landsat-8 images and has high accuracy. Based on the accuracy evaluation results after excluding the stations that cannot observe snow at all, the OA, PA, UA and CK of the AWT MODIS FSC product can reach 93.26%, 84.41%, 82.14% and 0.79, respectively. The AWT MODIS FSC product consists of two parts: the retrieval results of MODIS clear sky observations and the spatiotemporal reconstruction results based on the MSTI algorithm. Snow depth observations from meteorological stations are also used to evaluate these two parts. The binary precision metrics of fractional snow cover based

on MODIS clear-sky observations are as follows: OA (95.36%), PA (87.75%), UA (86.86%) and CK (0.84). The binarization accuracy metrics of the fractional snow cover based on the spatiotemporal reconstruction of the MSTI algorithm are as follows: OA (88.96%), PA (82.26%), UA (78.86%), CK (0.72).Therefore, it can be shown that both the binarized identification and fractional snow cover metrics are excellent at both the point scale and the areal scale, which further indicates that this AWT MODIS FSC product has relatively high precision(Wu et al., 2021; Huang et al., 2022; Hao et al., 2022). The AWT MODIS

FSC product is expected to offer robust and highly accurate data support for future snow hydrology studies.

## Author contribution

PFB performed the study and wrote the manuscript. JLM conceived the study and supervised the manuscript construction and revision. WGX, PJM, HJY, ZC, CHZ, and YJW helped in algorithm development and data processing. ZZJ provided the meteorological station data. WSL and SJC provided many meaningful suggestions. All authors have read and agreed to the

published version of the manuscript.

## Competing interests

The authors declare that they have no conflicts of interest.



## Acknowledgement

The authors would like to thank the National Meteorological Information Center (NMIC) for the weather station data. We also
acknowledge that the Google Earth Engine dramatically facilitated the work on image reprocessing.

## Financial support

This research was funded by the Second Tibetan Plateau Scientific Expedition and Research Program (STEP) under Grant No.
2019QZKK0206, the Strategic Priority Research Program of Chinese Academy of Sciences (XDA20100300) and the National
Natural Science Foundation of China (42171317, 42090014).

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
