# Peer review of "MODIS Daily Cloud-gap-filled Fractional Snow Cover Dataset of the Asian Water Tower Region (2000-2022)"

_Earth System Science Data, 2023_

## Author Comment (AC1)

**Response to Reviewer 2**

Pan's paper produced a daily cloud-free daily Cloud-gap-filled fractional snow cover products with 0.005°spatial resolution based on MODIS surface reflectance data in the Asia Water Tower (AWT) region. The core of the algorithm contains two parts, one part is the automatic endmember extraction (MESMA-AGE) technique for producing FSC, and the other part is the multistep spatiotemporal interpolation algorithm (MSTI) for filling gaps. However, before this article is accepted by ESSD, the following questions are still required.

We thank the topic editor, editor and anonymous reviewers for their thoughtful and constructive comments and suggestions, which significantly help us to improve the quality of the manuscript. In this revised manuscript, we have tried our best as much as possible to address all concerns and have revised the manuscript accordingly. Below, we indicate the original comment of the reviewer in black and our point-to-point response is denoted in blue.

**General comments:**

1. The author conducted a comprehensive evaluation of the product using Landsat8 and sites, but comparison with existing products was lacking. It is recommended to increase comparison with other 500-meter fractional snow cover products. The author also mentioned that SNOW CCI data can be compared before going to the cloud to clarify the accuracy of the MESMA-AGE algorithm. In particular, the MOD10A1F product is global daily Cloud-gap-filled fractional snow cover dataset. This study does not mention or compare with it.

**Response:** We greatly thank the Reviewer #2 for the comment. Following the reviewer's suggestion, we added the comparison with the Snow CCI(MODIS) products on the continuous value accuracy evaluation index in the clear sky scenario. The result is shown in Figure 1 below. As shown in Figure 1, the $R^2$, RMSE and MAE of the AWT MODIS FSC product are 0.831, 0.148 and 0.084. The $R^2$, RMSE and MAE of the Snow CCI(MODIS) product are 0.780, 0.159 and 0.094, respectively. It can be seen from Figure1 that the products we have produced are overall better than the Snow CCI(MODIS) products. We have also added the relevant results to **Section 5.1**.

[Figure]

**Figure 1.** Violin charts for the accuracy evaluation metrics of the AWT MODIS FSC and Snow CCI(MODIS) product validated by Landsat images

We have compared MOD10A1 products, and it can be seen from the relevant results in **Section 5.1** that our AWT MODIS FSC products are superior to MOD10A1 under clear sky conditions no matter how NDSI threshold is selected. And previous studies have shown that the MOD10A1F product have a

lower verified accuracy than MOD10A1 products (Hao et al., 2022; Stillinger et al., 2023). Because it's based on the MOD10A1 product, their cloud gap filled (CGF) algorithm virtually only replaces cloud gaps in the current day with the previous most-recent clear-sky observation (Hall et al., 2019). Meanwhile, this CGF algorithm fails to consider the advantages of Terra and Aqua MODIS cooperative observations and the surrounding spatio-temporal information, resulting in limited accuracy of the interpolation results. Therefore, this study did not compare with MOD10A1F products. However, as suggested by the reviewer, we have added the introduction of the MOD10A1F products in the **Section 1**.

**Reference:**

Hall, D. K., G. A. Riggs, N.E. DiGirolamo and M.O. Román, 2019: MODIS Cloud-Gap Filled Snow-Cover Products: Advantages and Uncertainties, Hydrology and Earth System Sciences, 23:5227-5241, https://doi.org/10.5194/hess-23-5227-2019.

Hao, X., Huang, G., Zheng, Z., Sun, X., Ji, W., Zhao, H., Wang, J., Li, H., and Wang, X.: Development and validation of a new MODIS snow-cover-extent product over China, Hydrol. Earth Syst. Sci., 26, 1937–1952, https://doi.org/10.5194/hess-26-1937-2022, 2022.

Stillinger, T., Rittger, K., Raleigh, M., Michell, A., Davis, R., and Bair, E.: Landsat, MODIS, and VIIRS snow cover mapping algorithm performance as validated by airborne lidar datasets, The Cryosphere, 17, 567–590, https://doi.org/10.5194/tc-17-567-2023, 2023.

2. The selection of pure end-member is the key to the MESMA-AGE algorithm. What is the basis for selecting pure end-member in TABLE2? Qinghai-Tibet soil and rocks are quite different. Is the current selection very representative? Further explanation is needed.

**Response:** We sincerely appreciate the feedback from Reviewer #2. Following the reviewer's suggestion, we have added reference citations in Table 2. The MESMA-AGE algorithm was specifically tailored to the study area. Pure snow samples were selected for training according to snow spectral curve and image features, and the snow extraction rules of NDSI greater than 0.7 and NDVI less than -0.035 were obtained. In addition, a criterion of green band greater than 0.7 was included to mitigate the effect of glacial lakes (Shi, 2012). Non-snow endmembers, which include vegetation, water bodies and bare soil/rock, were identified primarily based on NDVI and spectral characteristics of non-snow endmembers (Shi, 2012). Simultaneously, the spectral vector algorithm (Xu et al., 2015) was employed to refine and optimize the spectral library, which obtained from images by using endmembers extraction rules (Shi, 2012). This refinement aimed to enhance the representativeness of the endmembers while ensuring computational efficiency. Comparative analysis with MODSCAG and MOD10A1 products revealed that the FSC retrieval results obtained through the MESMA-AGE algorithm exhibited superior accuracy overall within our study area (Hao et al., 2019). This comparison underscores the validity and reliability of the endmember extraction criteria outlined in Table 2. Considering that if the image area is too large, the representativeness will be limited if only one set of endmember libraries is used, this study performs the FSC retrieval independently for each MODIS tile during the retrieval process. To further validate the reliability of the endmember extraction rule, this study expanded the samples depicted in Figure 3 of the article. A total of 42,033 samples were selected from January, June, November, and December of 2001,

2005, 2010, 2015, and 2020, including 14756 snow samples 6968, vegetation samples and 20309 soil samples. The soil/rocks on the Qinghai-Tibet Plateau do possess unique characteristics, as mentioned by the reviewer. Specifically, the high salt content in certain areas and surface brightness contribute to certain misinterpretations in the retrieval results. Consequently, to better demonstrate the efficacy of the soil endmember extraction rules, we intentionally augmented the number of soil samples. These samples were utilized to construct the ground object feature map (Figure 2), where the x-axis represents the NDSI value, and the y-axis denotes the NDVI. The endmember extraction rules outlined in Table 2 effectively identify and isolate regions located at the geometric vertices within the two-dimensional scatter plots in Figure 2.

**Therefore, we modify the corresponding statement in Section 3.1: "In this study, the MEAMA-AGE algorithm was used to retrieve the fractional snow cover in the Asian Water Tower region, which combines an image-based automatic endmember extraction algorithm(Shi, 2012) with a spectral library optimization method (Xu et al., 2015). The MESMA-AGE algorithm can improve the computational efficiency while ensuring the representativeness of the endmembers(Hao et al., 2019). Considering that if the image area is too large, the representativeness will be limited if only one set of endmember libraries is used, this study performs the FSC retrieval independently for each MODIS tile during the retrieval process. The rules for extracting snow and non-snow endmembers are shown in Table 2.**

**For this study, 42033 samples were selected from the MODIS data in January, November, and December 2020. The sample types include snow (14756 samples), vegetation (6968 samples), and soil (20309 samples). The samples were used to create the ground object feature map (Fig. 3). The x-axis in Fig. 3 is the NDSI value, and the y-axis is the NDVI. The endmember extraction rules outlined in Table 2 effectively identify and isolate regions located at the geometric vertices within the two-dimensional scatter plots in Figure 3."**

[Figure]

Figure 2. The NDSI and NDVI pattern of vegetation, soil/rock, and snow endmembers from MODIS images

3. In 3.2 "Multistep spatiotemporal interpolation algorithm", the author's last two steps are "piecewise cubic Hermite interpolating polynomial (PCHIP) for the 19-day period, and further spatial interpolation

using a 10×10 window." However, the snow cover on the Tibetan Plateau changes rapidly and has strong spatial heterogeneity. The author also mentioned in "4.2.3 Performance of spatiotemporal reconstruction algorithm" that "The results show that the accuracy of the fractional snow cover based on clear sky observations is significantly better than that of the spatiotemporal reconstruction. "The 19-day time series and the 10 ×10 window" interpolation may not be suitable for the Tibetan Plateau, and the author needs to further consider the rationality of the Multistep spatiotemporal interpolation algorithm.

**Response:** We greatly thank the Reviewer #2 for the comment. Although the accuracy of spatio-temporal interpolation using more recent spatio-temporal information is higher, the clouds in the AWT region have the characteristics of wide coverage and long duration. In this study, after performing the first and second steps of the MSTI algorithm (temporal information of the surrounding front and back days and spatial information of the surrounding neighboring pixels), it is found that there is still a high number of cloud-day. Therefore, the cloud persistence days (CPD) of each cloud pixel is calculated based on the daily MOD09GA/MYD09GA combination image during 2000–2019 and the results are shown in Figure 3. From the figure, only 3.42% of the remaining proportion of CPD is greater than 20 days, so this study chooses 19 days as the time window and interpolates it using the PCHIP algorithm. Combined with Figure 17(a) in the revised manuscript, the regions where the cloud exists for a long time and over a wide area are mainly the regions with relatively stable snow cover, such as the Pamir Plateau, the Himalayan Mountains, the Altay region and the Hengduan Mountains, or the regions with almost no snow in the south of the Himalayas. Therefore, although the time window is longer, the high accuracy can still be guaranteed in these regions. Specifically, we can see the accuracy evaluation results of the mountainous area in Figure 6 in the revised manuscript. Meanwhile, compared with the spline interpolation algorithm used in existing studies (Dozier et al., 2008; Tang et al., 2013, 2022), which uses the whole sequence information to fit an equation, the PCHIP algorithm (Fritsch and Carlson., 1980) divides the time series into several sub-intervals, and the fitting equation for this sub-interval can be obtained only by using the two endpoints of the sub-intervals and their derivative values. This also can make the results more conformal since the adjacent sub-intervals share an endpoint and a derivative. For details about the PCHIP algorithm, please see Formulas 5-8 in Section 3.2. Therefore, the PCHIP algorithm can adaptively select a suitable time window for interpolation according to the CPD, which ensures the monotonicity of the interpolation result, thus suppressing the influence of noise while achieving a spatio-temporally continuous snow cover and avoiding results outside the reasonable range of the snow cover. Although the PCHIP algorithm has introduced some errors, it is still better than the spline interpolation algorithm used in the previous study. High-frequency observations from geostationary meteorological satellites provide more opportunity to eliminate the influence of clouds on the extraction of snow information. Next work we will consider the combination of geostationary satellites in snow cover mapping. We have also made changes in section 3.2 of the article to give readers a better understanding of the PCHIP algorithm's ability to adapt to select appropriate sub-windows based on CPD within a 19-day window.

[Figure]

Figure 3. The mean frequency of CPD during 2000–2019

We strongly agree with the reviewer's suggestion that the snow cover in the Asian Water Tower region has strong spatial heterogeneity. Therefore, in the fourth step of the MSTI algorithm, we adopt the inverse distance weight interpolation algorithm considering the elevation information, which can exclude pixels with large spatial heterogeneity with the elevation information, and adjust the weight of pixels with different distances by using the inverse distance weight algorithm. Through extensive testing, we found that using the 11*11 window can fill all the remaining cloud-covered pixels in the first three steps.

**Reference:**

Dozier, J., Painter, T. H., Rittger, K., and Frew, J. E.: Time–space continuity of daily maps of fractional snow cover and albedo from MODIS, Adv. Water Resour., 31, 1515–1526, https://doi.org/10.1016/j.advwatres.2008.08.011, 2008.

Fritsch, F. N. and Carlson, R. E.: Monotone Piecewise Cubic Interpolation, SIAM J. Numer. Anal., 17, 238–246, https://doi.org/10.1137/0717021, 1980.

Tang, Z., Wang, J., Li, H., and Yan, L.: Spatiotemporal changes of snow cover over the Tibetan plateau based on cloud-removed moderate resolution imaging spectroradiometer fractional snow cover product from 2001 to 2011, J. Appl. Remote Sens., 7, 073582, https://doi.org/10.1117/1.JRS.7.073582, 2013.

Tang, Z., Deng, G., Hu, G., Zhang, H., Pan, H., and Sang, G.: Satellite observed spatiotemporal variability of snow cover and snow phenology over high mountain Asia from 2002 to 2021, J. Hydrol., 613, 128438, https://doi.org/10.1016/j.jhydrol.2022.128438, 2022.

4.    In previous studies, FSC set thresholds greater than 10, 15... to consider snow, and site verification set thresholds greater than 0cm, 1cm, 2cm, 3, cm, 4cm, 5cm... both, this study Please explain further the reasons for using FSC>15 and sd>3cm.

**Response:** We greatly thank the Reviewer #2 for the comment. We refer to the value of the FSC threshold in the verification process of the MODSCAG algorithm during the product validation (Painter et al., 2009). The MODSCAG algorithm validation process suggests that the error for FSC less than 15% would probably be larger because the spectral signal from snow is diminished with mixing from other land covers (Painter et al., 2009; Rittger et al., 2013) and cloud/snow misjudgments are more severe in areas with less snow (Hall and Riggs, 2007; Tang et al., 2013). And this threshold has been widely used in the

validation of VIIRSCAG (Rittger et al., 2021), USGS Landsat fSCA (Selkowitz et al., 2017), and GOESRSCAG (Key et al., 2020) products. Therefore, in this study, 15% was used as the threshold for binary discrimination of snow. **The FSC threshold is further explained in Section 3.3: " To calculate the overall accuracy, binary snow cover was labeled for pixels with fractional snow cover ≥15%, as the error for FSC less than 15% would probably be larger because the spectral signal from snow is diminished with mixing from other land covers and cloud/snow misjudgments are more severe in areas with less snow (Painter et al., 2009; Rittger et al., 2013, 2021; Selkowitz et al., 2017; Key et al., 2020; Hall and Riggs, 2007; Tang et al., 2013a)."**

This study chose 3 cm as the threshold for binary snow identification of snow depth data based on the previous studies on the verification of snow cover on the Tibetan Plateau (Yang et al., 2015; Zhang et al., 2019; Huang et al., 2022a, b). These studies suggest that the small snow depth should considered as trace introducing large uncertainties possibly due to more susceptibility to the time difference between satellite and ground observations, more patchy vegetation, higher possibility of erroneously classifying thin snow as clouds (Ault et al., 2006; Ke et al., 2016; Zhang et al., 2019; Wang et al., 2022). And due to the fragmentation of snow in the Tibetan Plateau, the snow depth of 3 cm can better ensure the snow coverage in the pixel, and can also better explain the accuracy of the snow binary products. At the same time, in the process of using snow depth to evaluate the accuracy of binary snow identification, FSC<15% of the pixels were reclassified as non-snow, so this study chose 3 cm as the threshold for binary snow identification of snow depth data based on the existing research. **The Snow Depth threshold is further explained in Section 2.4: "Snow depth data can only be used to evaluate binarized snow products, whereas the AWT MODIS FSC products are binarized by re-classifying the image pixels with small fractional snow cover as no snow, and smaller snow depths tend to have greater uncertainty (Ault et al., 2006; Ke et al., 2016; Zhang et al., 2019; Wang et al., 2022). Therefore, this study refers to previous studies to binaries the snow depth data with a threshold of 3 cm in AWT region (Yang et al., 2015; Huang et al., 2022a, b; Zhang et al., 2019), i.e. snow depths less than 3 cm are classified as no snow and those greater than 3 cm are classified as snow."**

**Reference:**

Ault, T. W., Czajkowski, K. P., Benko, T., Coss, J., Struble, J., Spongberg, A., Templin, M., and Gross, C.: Validation of the MODIS snow product and cloud mask using student and NWS cooperative station observations in the Lower Great Lakes Region, Remote Sens. Environ., 105, 341–353, https://doi.org/10.1016/j.rse.2006.07.004, 2006.

Hall, D. K. and Riggs, G. A.: Accuracy assessment of the MODIS snow products, Hydrol. Process., 21, 1534–1547, https://doi.org/10.1002/hyp.6715, 2007.

Huang, Y., Song, Z., Yang, H., Yu, B., Liu, H., Che, T., Chen, J., Wu, J., Shu, S., Peng, X., Zheng, Z., and Xu, J.: Snow cover detection in mid-latitude mountainous and polar regions using nighttime light data, Remote Sens. Environ., 268, 112766, https://doi.org/10.1016/j.rse.2021.112766, 2022a.

Huang, Y., Xu, J., Xu, J., Zhao, Y., Yu, B., Liu, H., Wang, S., Xu, W., Wu, J., and Zheng, Z.: HMRFS-TP: long-term daily gap-free snow cover products over the Tibetan Plateau from 2002 to 2021 based on Hidden Markov Random Field model, Snow and Sea Ice, https://doi.org/10.5194/essd-2022-134, 2022b.

Ke, C.-Q., Li, X.-C., Xie, H., Ma, D.-H., Liu, X., and Kou, C.: Variability in snow cover phenology in China from 1952 to 2010, Hydrol. Earth Syst. Sci., 20, 755–770, https://doi.org/10.5194/hess-20-755-2016, 2016.

Key, J., Liu, Y., Wang, X., Letterly, A., and Painter, T.: Snow and Ice Products from ABI on the GOES-R Series, 165–177, https://doi.org/10.1016/B978-0-12-814327-8.00014-7, 2020.

Painter, T. H., Rittger, K., McKenzie, C., Slaughter, P., Davis, R. E., and Dozier, J.: Retrieval of subpixel snow covered area, grain size, and albedo from MODIS, Remote Sens. Environ., 113, 868–879, https://doi.org/10.1016/j.rse.2009.01.001, 2009.

Rittger, K., Painter, T. H., and Dozier, J.: Assessment of methods for mapping snow cover from MODIS, Adv. Water Resour., 51, 367–380, https://doi.org/10.1016/j.advwatres.2012.03.002, 2013.

Rittger, K., Bormann, K. J., Bair, E. H., Dozier, J., and Painter, T. H.: Evaluation of VIIRS and MODIS Snow Cover Fraction in High-Mountain Asia Using Landsat 8 OLI, Front. Remote Sens., 2, 2021.

Selkowitz, D. J., Painter, T. H., Rittger, K. E., Schmidt, G., and Forster, R.: The USGS Landsat Snow Covered Area Products: Methods and Preliminary Validation, in: Automated Approaches for Snow and Ice Cover Monitoring Using Optical Remote Sensing, edited by: Selkowitz, D. J., The University of Utah, Salt Lake City, Utah, 76–119, 2017.

Tang, B.-H., Shrestha, B., Li, Z.-L., Liu, G., Ouyang, H., Gurung, D. R., Giriraj, A., and Aung, K. S.: Determination of snow cover from MODIS data for the Tibetan Plateau region, Int. J. Appl. Earth Obs. Geoinformation, 21, 356–365, https://doi.org/10.1016/j.jag.2012.07.014, 2013.

Wang, G., Jiang, L., Xiong, C., and Zhang, Y.: Characterization of NDSI Variation: Implications for Snow Cover Mapping, IEEE Trans. Geosci. Remote Sens., 60, 1–18, https://doi.org/10.1109/TGRS.2022.3165986, 2022.

Yang, J., Jiang, L., Menard, C., Luojus, K., Lemmetyinen, J., and Pulliainen, J.: Evaluation of snow products over the Tibetan Plateau, Hydrol. Process., 29, https://doi.org/10.1002/hyp.10427, 2015.

Zhang, H., Zhang, F., Zhang, G., Che, T., Yan, W., Ye, M., and Ma, N.: Ground-based evaluation of MODIS snow cover product V6 across China: Implications for the selection of NDSI threshold, Sci. Total Environ., 651, 2712–2726, https://doi.org/10.1016/j.scitotenv.2018.10.128, 2019.

5.  It is recommended that the author add a schematic diagram of Asian Water Tower Region's products so that readers can more intuitively understand the algorithm effect of each step.

**Response:** We greatly thank the Reviewer #2 for the comment. According to the suggestions of reviewers, we modified the flow chart to make it easier for readers to understand the whole FSC retrieval and spatio-temporal interpolation process and the algorithm effect of each step.

[Figure]

Figure 4. Overall flowchart of the AWT MODIS FSC product

**Minor comments:**

1. There are two RMSE indicators in Figure 2. It is recommended to modify one of them to MAE.

**Response:** We greatly thank the Reviewer #2 for the comment. Sorry for the error caused by our carelessness, we have modified the figure 2.

2. Line 207 mentioned "further spatial interpolation using a 10×10 window", and Line 241 mentioned "In this study, the observation information from the 11*11 interpolation window centered on the cloud pixel was used based on the inverse distance weight ", did the author use "10×10" or "11×11" window in the fourth step of the multistep spatiotemporal interpolation algorithm?

**Response:** We greatly thank the Reviewer #2 for the comment. Sorry for the misunderstanding caused by our careless, we use the information in the 11*11 interpolation window centred on the cloud pixel to perform spatial interpolation. **We have revised the corresponding Line 207 expression: "further spatial interpolation using a 11×11 window".**

---

## Author Comment (AC2)

**Response to Reviewer #1**

We thank the topic editor, editor and anonymous reviewers for their thoughtful and constructive comments and suggestions, which significantly help us to improve the quality of the manuscript. In this revised manuscript, we have tried our best as much as possible to address all concerns and have revised the manuscript accordingly. Below, we indicate the original comment of the reviewer in black and our point-to-point response is denoted in blue.

1. Please add a reference for Table 2, and justify why the thresholds are reasonable and reliable in the study area.

**Response:** We sincerely appreciate the feedback from Reviewer #1. Following the reviewer's suggestion, we have added reference citations to Table 2. The MESMA-AGE algorithm was specifically tailored to the study area. Pure snow samples were selected for training according to snow spectral curve and image features, and the snow extraction rules of NDSI greater than 0.7 and NDVI less than -0.035 were obtained. In addition, a criterion of green band greater than 0.7 was included to mitigate the effect of glacial lakes (Shi, 2012). Non-snow endmembers, which include vegetation, water bodies and bare soil/rock, were identified primarily based on NDVI and spectral characteristics of non-snow endmembers (Shi, 2012). Simultaneously, the spectral vector algorithm (Xu et al., 2015) was employed to refine and optimize the spectral library, which obtained from images by using endmembers extraction rules (Shi, 2012). This refinement aimed to enhance the representativeness of the endmembers while ensuring computational efficiency. Comparative analysis with MODSCAG and MOD10A1 products revealed that the FSC retrieval results obtained through the MESMA-AGE algorithm exhibited superior accuracy overall within our study area (Hao et al., 2019). This comparison underscores the validity and reliability of the endmember extraction criteria outlined in Table 2. Considering that if the image area is too large, the representativeness will be limited if only one set of endmember libraries is used, this study performs the FSC retrieval independently for each MODIS tile during the retrieval process. To further validate the reliability of the endmember extraction rule, this study expanded the samples depicted in Figure 3 of the article. A total of 42,033 samples were selected from January, June, November, and December of 2001, 2005, 2010, 2015, and 2020, including 14756 snow samples 6968, vegetation samples and 20309 soil samples. These samples were utilized to construct the ground object feature map (Figure 1), where the x-axis represents the NDSI value, and the y-axis denotes the NDVI. The endmember extraction rules outlined in Table 2 effectively identify and isolate regions located at the geometric vertices within the two-dimensional scatter plots in Figure 1.

**Therefore, we modify the corresponding statement in Section 3.1: "In this study, the MEAMA-AGE algorithm was used to retrieve the fractional snow cover in the Asian Water Tower region, which combines an image-based automatic endmember extraction algorithm (Shi, 2012) with a spectral library optimization method (Xu et al., 2015). The MESMA-AGE algorithm can improve the computational efficiency while ensuring the representativeness of the endmembers (Hao et al., 2019). Considering that if the image area is too large, the representativeness will be limited if only one set of endmember libraries is used, this study performs the FSC retrieval**

**independently for each MODIS tile during the retrieval process. The rules for extracting snow and non-snow endmembers are shown in Table 2.**

**For this study, 42033 samples were selected from the MODIS data in January, November, and December 2020. The sample types include snow (14756 samples), vegetation (6968 samples), and soil (20309 samples). The samples were used to create the ground object feature map (Fig. 3). The x-axis in Fig. 3 is the NDSI value, and the y-axis is the NDVI. The endmember extraction rules outlined in Table 2 effectively identify and isolate regions located at the geometric vertices within the two-dimensional scatter plots in Figure 3."**

[Figure]

Figure 1. The NDSI and NDVI pattern of vegetation, soil/rock, and snow endmembers from MODIS images

**Reference:**

Hao, S., Jiang, L., Shi, J., Wang, G., and Liu, X.: Assessment of MODIS-Based Fractional Snow Cover Products Over the Tibetan Plateau, IEEE J. Sel. Top. Appl. Earth Obs. Remote Sens., 12, 533–548, https://doi.org/10.1109/JSTARS.2018.2879666, 2019.

Shi, J.: An Automatic Algorithm on Estimating Sub-Pixel Snow Cover from MODIS, Quat. Sci., 32, 6–15, 2012.

Xu, Y., Shi, J., and Du, J.: An Improved Endmember Selection Method Based on Vector Length for MODIS Reflectance Channels, Remote Sens., 7, 6280–6295, https://doi.org/10.3390/rs70506280, 2015.

2. Figure 3. It is true that the NDSI and NDVI of vegetation, soil, and snow endmembers show much difference. However, I am curious about whether the approach works well for some vertically mixed pixels. For example, forest or shrub-covered snow represent a special mixture of spectral information. Please discuss the performance of the algorithm and products in these regions. Some studies have suggested that NDSI does not show many changes with or without snow under the canopy of boreal forest.

**Response:** We greatly thank the Reviewer #1 for the comment. As mentioned by the reviewer, the presence of vegetation canopy can indeed interfere with the accurate observation of snow signals, leading to inaccuracies in the FSC retrieval results. Our team also used the GORT model to simulate NDSI changes under different forest canopies (Wang et al., 2022), as shown in Figure 2 below. As can be seen in Figure 2, as mentioned by the reviewer in forest areas with relatively high forest cover NDSI does not show much variation with or without snow. Therefore, the accuracy of algorithms and products that use

NDSI and FSC to establish fitting relationships is limited. However, the MESMA-AGE algorithm employed in this study dynamically extracted ground-object endmembers using strict criteria, thus ensuring the model's capability to precisely differentiate between snow and vegetation. The relevant results are shown in Figure 6 in subsection 4.1.2 of the revised manuscript. It can be seen from the results that the accuracy of our products in forest areas is like that in other surface types over Asian Water Tower. On the other hand, our team has previously compared our algorithm with the MODSCAG and MOD1010A1 products in forested areas (Hao et al., 2019). Although the MODSCAG product is adjusted for canopy (Raleigh et al., 2013), the results show that there is some overestimation of the MODSCAG product in forested gaps. This because canopy adjustment method assumes that the FSC under the canopy matches the FSC in the areas of canopy gaps. However, the snow accumulation and ablation processes are not identical for under canopy snow and canopy-free snow or for various forest types (Varhola et al., 2010). Snow cover in viewable areas is not always suitable to represent the snow cover under the canopy. At the same time, as can be seen from the surface classification in Figure 1 of the revised manuscript, there is less forest snow cover and more bare snow cover in the AWT region, so we have not applied a canopy adjustment. However, if users are interested, they can adjust it using readily available FVC products. And in areas of particularly high FVC, we suggest that URSI or NDFSI, which are more sensitive to vegetation, can be used to replace NDSI for greater accuracy. **Therefore, we modify the corresponding statement in Section 4.1.2: "From the perspective of fractional snow cover accuracy metrics, grassland and forest are slightly worse, mainly because it is difficult to observe the snow signal shielded by the vegetation canopy at the MODIS scale. Previous studies have demonstrated that canopy adjustment using fractional vegetation cover (FVC) can enhance the accuracy of observations in such areas (Raleigh et al., 2013; Rittger et al., 2020; Xiao et al., 2022). Therefore, future relevant studies can utilize mature FVC products for canopy adjustment to fulfill research requirements. For snow mapping in areas with high forest cover we recommend using URSI (Wang et al., 2021) or NDFSI (Wang et al., 2020), which are more sensitive indicators, to replace NDSI to ensure accuracy."**

[Figure]

Figure 2. Relationships between the NDSI in the Principal plane (PP) and $1 - F_{canopy}$ derived from the GORT model for various illuminating-viewing geometries and equivalent grain sizes (EGSs) (Wang et al., 2022)

**Reference:**

Hao, S., Jiang, L., Shi, J., Wang, G., and Liu, X.: Assessment of MODIS-Based Fractional Snow Cover Products Over the Tibetan Plateau, IEEE J. Sel. Top. Appl. Earth Obs. Remote Sens., 12, 533–548, https://doi.org/10.1109/JSTARS.2018.2879666, 2019.

Raleigh, M. S., Rittger, K., Moore, C. E., Henn, B., J. Lutz, A., and Lundquist, J. D. Ground-based testing of MODIS fractional snow cover in subalpine meadows and forests of the Sierra Nevada, Remote Sens. Environ., vol. 128, pp. 44–57, 2013.

Rittger, K., Raleigh, M. S., Dozier, J., Hill, A. F., Lutz, J. A., and Painter, T. H.: Canopy Adjustment and Improved Cloud Detection for Remotely Sensed Snow Cover Mapping, Water Resour. Res., 56, e2019WR024914, https://doi.org/10.1029/2019WR024914, 2020.

Varhola, A., Coops, N. C., Weiler, M., and Moore, R. D. Forest canopy effects on snow accumulation and ablation: An integrative review of empirical results, J. Hydrol., vol. 392, no. 3, pp. 219–233, 2010.

Xiao, X., He, T., Liang, S., Liu, X., Ma, Y., Liang, S., and Chen, X.: Estimating fractional snow cover in vegetated environments using MODIS surface reflectance data, Int. J. Appl. Earth Obs. Geoinformation, 114, 103030, https://doi.org/10.1016/j.jag.2022.103030, 2022.

Wang, G., Jiang, L., Xiong, C., and Zhang, Y.: Characterization of NDSI Variation: Implications for Snow Cover Mapping, IEEE Trans. Geosci. Remote Sens., 60, 1–18, https://doi.org/10.1109/TGRS.2022.3165986, 2022.

Wang, G., Jiang, L., Shi, J., and Su, X.: A Universal Ratio Snow Index for Fractional Snow Cover Estimation, IEEE Geosci. Remote Sens. Lett., 18, 721–725, https://doi.org/10.1109/LGRS.2020.2982053, 2021.

Wang, X., Chen, S., and Wang, J.: An Adaptive Snow Identification Algorithm in the Forests of Northeast China, IEEE J. Sel. Top. Appl. Earth Obs. Remote Sens., 13, 5211–5222, https://doi.org/10.1109/JSTARS.2020.3020168, 2020.

3. L212-214, L230-232. The authors indicated that the snow cover in the area change rapidly, and thus selected a small time window for the temporal interpolation (3 day). However, in the following step, they applied a 9-day window to remove the rest cloud pixels (PCHIP method). It does not make sense to me. Did this PCHIP approach brings too much errors?

**Response:** We greatly thank the Reviewer #1 for the comment. As mentioned by the reviewer, the shorter the time window, the higher the accuracy, but only two points of the data before and after the cloud day cannot be used by the PCHIP algorithm. So we first use the clear sky average value before and after the cloud day to fill in. This ensures accuracy, and the overall operation efficiency can be greatly improved because the PCHIP algorithm is very time consuming. The clouds in the AWT region have the characteristics of wide coverage and long duration, and after performing the first and second steps of the MSTI algorithm (temporal information of the front and back days and spatial information of the surrounding neighboring pixels), it is found that there is still a high number of cloud-day. Therefore, the cloud persistence days (CPD) of each cloud pixel is calculated based on the daily MOD09GA/MYD09GA combination image during 2001–2020 and the results are shown in Figure 2. From the figure, only 3.42% of the remaining proportion of CPD is greater than 20 days, so this study chooses 19 days as the time window and interpolates it using the PCHIP algorithm. Combined with Figure 17(a) in the revised manuscript, the regions where the cloud exists for a long time and over a wide area are mainly the regions with relatively stable snow cover, such as the Pamir Plateau, the Himalayan Mountains, the Altay region and the Hengduan Mountains, or the regions with almost no snow in the

south of the Himalayas. Therefore, although the time window is longer, the high accuracy can still be guaranteed in these regions. Specifically, we can see the accuracy evaluation results of the mountainous area in Figure 6 of the revised manuscript. Meanwhile, compared with the spline interpolation algorithm used in existing studies (Dozier et al., 2008; Tang et al., 2013, 2022), which uses the whole sequence information to fit an equation, the PCHIP algorithm (Fritsch and Carlson., 1980) divides the time series into several sub-intervals, and the fitting equation for this sub-interval can be obtained only by using the two endpoints of the sub-intervals and their derivative values. This also can make the results more conformal since the adjacent sub-intervals share an endpoint and a derivative. For details about the PCHIP algorithm, please see Formulas 5-8 in Section 3.2. Therefore, the PCHIP algorithm can adaptively select a suitable time window for interpolation according to the CPD, which ensures the monotonicity of the interpolation result, thus suppressing the influence of noise while achieving a spatio-temporally continuous snow cover and avoiding results outside the reasonable range of the snow cover. Although the PCHIP algorithm has introduced some errors, it is still better than the Cubic algorithm used in the previous study. High-frequency observations from geostationary meteorological satellites provide more opportunity to eliminate the influence of clouds on the extraction of snow information. Next work we will consider the combination of geostationary satellites in snow cover mapping. We have also made changes in section 3.2 of the article to give readers a better understanding of the PCHIP algorithm's ability to adapt to select appropriate sub-windows based on CPD within a 19-day window.

[Figure]

Figure 2. The mean frequency of CPD during 2001–2020

**References:**

Dozier, J., Painter, T. H., Rittger, K., and Frew, J. E.: Time–space continuity of daily maps of fractional snow cover and albedo from MODIS, Adv. Water Resour., 31, 1515–1526, https://doi.org/10.1016/j.advwatres.2008.08.011, 2008.

Fritsch, F. N. and Carlson, R. E.: Monotone Piecewise Cubic Interpolation, SIAM J. Numer. Anal., 17, 238–246, https://doi.org/10.1137/0717021, 1980.

Tang, Z., Wang, J., Li, H., and Yan, L.: Spatiotemporal changes of snow cover over the Tibetan plateau based on cloud-removed moderate resolution imaging spectroradiometer fractional snow cover product from 2001 to 2011, J. Appl. Remote Sens., 7, 073582, https://doi.org/10.1117/1.JRS.7.073582, 2013.

Tang, Z., Deng, G., Hu, G., Zhang, H., Pan, H., and Sang, G.: Satellite observed spatiotemporal variability of snow cover and snow phenology over high mountain Asia from 2002 to 2021, J. Hydrol., 613, 128438, https://doi.org/10.1016/j.jhydrol.2022.128438, 2022.

4. L363-365. The authors argued that increase in the amount of station data and observations lead to better accuracy assessment. It does not make sense. Were the station data only used for verification of the FSC products? Were they used for the training of the algorithm? If not, I think more station data for verification would not increase the accuracy of the developed products. Declined cloud cover should have contributed to the better verification as MODIS has higher accuracy in cloud-free days. I am not sure whether changes of snow cover days also affected the accuracy indices. It seems the high OA and CK values after 2015 were mainly driving by higher PA index. It likely means the omission errors decreased. Was there less snow cover in the AWT area in these years?

**Response:** We greatly thank the Reviewer #1 for the comment. Sorry for the misunderstanding caused by our expression. The snow depth data from the weather station was only used to evaluate the algorithms and products, not to train the algorithms. In this paper, we want to express that the decrease of cloud cover from 2000 to 2019 (see Figure 3 in this document, and Figure 9 in the revised manuscript) and the increase in the amount of observation data (see Figure 8 in the revised manuscript) have led to the **increase in the proportion of clear sky in the observation stations**. And this will lead to an increase in accuracy, especially after 2014. According to the suggestions of reviewers, we also calculated the annual mean fractional snow cover (FSC) and snow cover days (SCD) during 2000-2019, and the results are shown in Figure 4 below. It can be seen from the figure that both FSC and SCD show a decreasing trend, which also represents the decrease of snow cover in the AWT region. **Therefore, we modify the corresponding statement: "Combined with Figures 8 and 9, the decrease in cloud and snow cover (Tang et al., 2022; Yao et al., 2022) leads to an increase in the proportion of clear sky and non-snow observations at stations. This will result in fewer omission errors (PA increases), ultimately leading to better site assessment accuracy."**

[Figure]

Figure 3. The percentage of cloud cover from 2000 to 2019

[Figure]

Figure 4. Interannual variation of fractional snow cover (FSC) and snow cover days (SCD)

**References:**

Tang, Z., Deng, G., Hu, G., Zhang, H., Pan, H., and Sang, G.: Satellite observed spatiotemporal variability of snow cover and snow phenology over high mountain Asia from 2002 to 2021, J. Hydrol., 613, 128438, https://doi.org/10.1016/j.jhydrol.2022.128438, 2022.

Yao, T., Bolch, T., Chen, D., Gao, J., Immerzeel, W., Piao, S., Su, F., Thompson, L., Wada, Y., Wang, L., Wang, T., Wu, G., Xu, B., Yang, W., Zhang, G., and Zhao, P.: The imbalance of the Asian water tower, Nat. Rev. Earth Environ., 3, 618–632, https://doi.org/10.1038/s43017-022-00299-4, 2022.

5. L492-494. It is true that there are many cloud/snow confusion errors in MODIS data. Some researchers (e.g. Dong and Menzel, 2016, Journal of Hydrology; Remote Sensing of Environment) have conducted some research on this topic and developed some algorithm to remove overestimated (misclassified) snow pixels on MODIS snow maps using station data. It seems the proposed algorithm here did not consider this problem. It would be helpful to add some discussions about this.

**Response:** We greatly thank the Reviewer #1 for this meaningful suggestion. As suggested by the reviewer, the cloud and snow misclassification of MODIS snow products can be improved by combining station observations, which is a promising direction. But it requires enough densely distributed stations in the study area. From the distribution of stations in Figure 1 of the revised manuscript, the distribution of stations in the AWT region is sparse due to the complex terrain. It is mainly distributed in the east and north of the study area, while there are few stations in the high mountain areas of the Qinghai-Tibet Plateau where the snow cover is relatively concentrated, making it difficult to apply this method on a large scale in the study area. This is very important for the subsequent study of specific small areas, so the relevant discussion is supplemented in **Section 5.3: "At the same time, problems with MODIS cloud products, such as overestimation and confusion error between clouds and snow, can be effectively improved by combining station observations (Dong and Menzel, 2016a, b), but this requires enough dense stations in the study area. The applicability of this method is limited due to terrain constraints in the AWT region, and subsequent studies in specific small areas may be referred to further improve product quality. Long-scale synchronous and high-frequency observations from geostationary meteorological satellites can overcome the shortcomings of the above methods and meet the needs of large-scale applications."**

---

## Author Response (AR2)

**Response to Topic editor**

We thank the topic editor, editor and anonymous reviewers for their thoughtful and constructive comments and suggestions, which significantly help us to improve the quality of the manuscript. In this revised manuscript, we have tried our best as much as possible to address all concerns and have revised the manuscript accordingly. Below, we indicate the original comment of the Topic Editor in black and our point-to-point response is denoted in blue.

1. I am glad to share the great news that the revised paper show great improvement and is in good shape now. One reviewer suggests you to go over the entire paper and double check any possible grammar or typo (unfortunately he doesnt provide any examples), so please do so. We are expecting to publish the paper after this minor revisions.

**Response:** We sincerely appreciate the feedback from the Topic Editor. We have carefully reviewed the entire paper, double check for grammar and typo, and corrected them in the revised version.